# Hormone replacement therapy for postmenopausal atherosclerosis is offset by late age iron deposition

**Tianze Xu[1†], Jing Cai[1†], Lei Wang[1], Li Xu[2], Hongting Zhao[2], Fudi Wang[3], Esther G Meyron-Holtz[4], Fanis Missirlis[5], Tong Qiao[1]\*, Kuanyu Li[1,2]\***

[1]State Key Laboratory of Pharmaceutical Biotechnology, Department of Vascular Surgery, The Affiliated Drum Tower Hospital of Nanjing University Medical School, Nanjing, China; [2]Jiangsu Key Laboratory of Molecular Medicine, Medical School of Nanjing University, Nanjing, China; [3]The Second Affiliated Hospital, School of Public Health, State Key Laboratory of Experimental Hematology, Zhejiang University School of Medicine, Hangzhou, China; [4]Faculty of Biotechnology and Food Engineering, Technion Israel Institute of Technology, Haifa, Israel; [5]Department of Physiology, Biophysics and Neuroscience, Cinvestav, Mexico, Mexico

**\*For correspondence:**
qiaotong@nju.edu.cn (TQ);
likuanyu@nju.edu.cn (KL)

[†]These authors contributed equally to this work

**Competing interest:** The authors declare that no competing interests exist.

**Abstract** Postmenopausal atherosclerosis (AS) has been attributed to estrogen deficiency. However, the beneficial effect of hormone replacement therapy (HRT) is lost in late postmenopausal women with atherogenesis. We asked whether aging-related iron accumulation affects estrogen receptor α (ERα) expression, thus explaining HRT inefficacy. A negative correlation has been observed between aging-related systemic iron deposition and ERα expression in postmenopausal AS patients. In an ovariectomized *Apoe*[-/-] mouse model, estradiol treatment had contrasting effects on ERα expression in early versus late postmenopausal mice. ERα expression was inhibited by iron treatment in cell culture and iron-overloaded mice. Combined treatment with estradiol and iron further decreased ERα expression, and the latter effect was mediated by iron-regulated E3 ligase Mdm2. In line with these observations, cellular cholesterol efflux was reduced, and endothelial homeostasis was disrupted. Consequently, AS was aggravated. Accordingly, systemic iron chelation attenuated estradiol-triggered progressive AS in late postmenopausal mice. Thus, iron and estradiol together downregulate ERα through Mdm2-mediated proteolysis, providing a potential explanation for failures of HRT in late postmenopausal subjects with aging-related iron accumulation. This study suggests that immediate HRT after menopause, along with appropriate iron chelation, might provide benefits from AS.

## Editor's evaluation

These important research findings provide new insights into the biology of aging and support an important role for iron accumulation in post-menopausal women as a major reason why estrogen therapy is not as effective in preventing atherosclerosis as it is in the pre-menopausal state. The evidence supporting the conclusions is compelling using both animals models and human tissues. This work will be of broad interest to researchers and clinicians.

## Introduction

Atherosclerosis (AS) is the leading cause of cardiovascular disease-associated deaths worldwide (*Moss et al., 2019*). It has been well recognized that postmenopausal women confront an increasing

risk of AS owing to estrogen deficiency and disturbance of the estrogen receptor (ER) regulatory network (*Moss et al., 2019*). Nevertheless, the therapeutic effect of hormone replacement therapy (HRT) remains controversial. The Women's Health Initiative and the Heart and Estrogen/Progestin Replacement Study reported that the atheroprotective effect of HRT in late postmenopausal women (commonly ages over 65) is lost or even worsened (*Hlatky et al., 2002*; *Rocca et al., 2014*; *Rossouw et al., 2002*). The underlying mechanism may be triggered by aging, which reduces the protection afforded by estrogen, particularly estradiol ($E_2$).

In general, the atheroprotective effect of estrogen is attributed to its interaction with estrogen receptor α (ERα), which participates in foam cell formation and vascular remodeling (*Murphy, 2011*). ERα deletion is reported to induce adiposity and increase atherosclerotic lesion size since the promoters of lipid metabolism-related genes, such as *Tgm2*, *Apoe,* and *Abca1*, contain ERα-binding sites (*Ribas et al., 2011*). In addition, estrogen binds to ERα tethered to the plasma membrane, which can stimulate vasodilatation via an endothelial nitric oxide synthase (eNOS)-dependent pathway (*Gavin et al., 2009*; *Teoh et al., 2020*). Moreover, VEGF promotes angiogenesis and is transcriptionally regulated by the $E_2$-ERα complex (*Gu et al., 2018*). ERα expression decreases after menopause (*Gavin et al., 2009*; *Zhang et al., 2019*), suggesting the critical role of ERα in blood vessel function in healthy pre- and postmenopausal women. Despite the above, few studies have focused on the regulation of ERα in postmenopausal women on HRT.

Upon binding, estrogen functions as an ERα activator (*Lung et al., 2020*) and initiates ERα dimerization and translocation into the nucleus. Estrogen treatment in vitro leads to a marked increase in *Esr1* (gene that encodes ERα) mRNA through the binding of the $E_2$-ERα complex to estrogen-responsive elements (EREs) in the promoter region of *Esr1* (*Pinzone et al., 2004*). Estrogen binding to ERα also rapidly stimulates ubiquitination and proteasomal degradation of ERα (*Pinzone et al., 2004*). Ubiquitination-dependent ERα cycling on and off the ERE promoter sites to activate or prevent target gene transcription depends on the presence of estrogen at an adequate concentration (*Zhou and Slingerland, 2014*). Ubiquitin ligases that have been implicated in ERα regulation include BRCA1, MDM2, SKP1–CUL1–F-box S-phase kinase-associated protein 2 (SCF[SKP2]), and E6-associated protein (E6AP), all of which promote estrogen-induced transcriptional activity with cell-type selectivity (*Zhou and Slingerland, 2014*). Of these, MDM2 is a single-subunit RING finger E3 protein, which does not always inversely correlate with ERα levels in cancer cells (*Duong et al., 2007*). The relationship between MDM2 and ERα in other contexts and cell types remains elusive.

Estrogen also regulates systemic iron homeostasis through hepcidin, an antimicrobial peptide of 25 amino acids that mediates endocytosis and degradation of the ferrous iron exporter ferroportin 1 (Fpn1) (*Nemeth et al., 2004*). The dimeric $E_2$-ERα complex binds to an ERE site within the promoter of the hepcidin gene (*Hamp*) to inhibit its expression (*Hou et al., 2012*). Hence, $E_2$ elevates circulating iron to compensate for blood loss during menstruation (*Badenhorst et al., 2021*). However, $E_2$ declines by >90% after menopause, while systemic iron content increases slowly by steady iron uptake over the years. Serum ferritin in postmenopausal women increased by 2–3 times compared with premenopausal women (*Huang et al., 2013*). We and others have previously reported that iron overload is a risk factor for atherosclerosis (*Cai et al., 2020*; *Vinchi et al., 2020*). Our study relied on the proatherogenic *Apoe*[-/-] mouse model, which can be manipulated genetically to cause iron overload (*Cai et al., 2020*). We aimed to investigate the impact of aging-related iron accumulation on the therapeutic effect of HRT on AS and explore the underlying mechanisms. First, we corroborated on a new cohort of postmenopausal AS patients that a negative correlation exists between high concentrations of serum-iron and -ferritin and low expression of lesion ERα. We hypothesized that gradual iron accumulation in postmenopausal women could explain poor HRT efficacy when applied late. We used *Apoe* aging female mice or ovariectomized (OVX) young mice as estrogen-deficient AS models to examine ERα expression following HRT therapy and the genetic iron-overload in *Apoe*[-/-] background (*Apoe*[-/-]*Fpn1*[Lyz2/Lyz2]) to address the relation of iron metabolism and HRT efficacy. We found that ERα expression responded positively to E2 administration under low or normal iron conditions but negatively under high iron conditions. Our results offer a potential solution to the riddle of HRT outcomes in early and late postmenopausal women since only the latter group shows accelerated ERα proteolysis through iron-mediated upregulation of *Mdm2*.

**Table 1.** Clinical data of 20 patients with atherosclerosis (AS).

|  | EPM | LPM | p-value |
|---|---|---|---|
| Age | 59.3 ± 3.7 | 77.5 ± 4.5 | <0.0001 |
| Risk factors |  |  |  |
| Smoking history | 2 | 1 | 0.232 |
| Hypercholesterolemia | 4 | 8 | 1.000 |
| Hypertension | 7 | 8 | 0.334 |
| Coronary artery disease | 4 | 6 | 0.241 |
| Cerebral infarction | 4 | 5 | 0.548 |
| Cholesterol-lowering drug usage | 5 | 8 | 1.000 |
| Symptoms |  |  |  |
| Chest distress | 5 | 6 | 0.548 |
| Dizzy | 6 | 8 | 0.081 |
| Plaque type |  |  |  |
| Stable plaque | 5 | 3 | 0.207 |
| Vulnerable plaque | 5 | 7 | 0.207 |

EPM, early postmenopausal; LPM, late postmenopausal.

The online version of this article includes the following source data for table 1:

**Source data 1.** Patient clinical information.

## Results

### ERα protein abundance correlates inversely with systemic and local plaque iron content in the tested aging postmenopausal women with AS

To build the link between iron content and ERα levels in different postmenopausal stages, we collected 8 plaques and 20 blood samples from postmenopausal AS patients (*Table 1*), divided a half into early (55–65 years old, n = 4/10, plaque/blood samples) and another half into late (>66 years old, n = 4/10) postmenopausal groups (EPM and LPM), respectively. Tissue ferritin was evaluated by immunohistochemistry (IHC) (*Figure 1A*) and immunoblotting (*Figure 1B*), serum ferritin by ELISA (*Figure 1C*), tissue iron by DAB-enhanced Prussian blue staining (*Figure 1A*), and serum iron by ferrozine assays (*Figure 1D*). The results showed that ferritin and iron levels were significantly increased in plaques and serum in the LPM compared to that in the EPM (*Figure 1A–D*). Consistently, serum hepcidin levels increased in the LPM group (*Figure 1E*). By contrast, plaque ERα expression was lower in the LPM than in the EPM (*Figure 1B*). In addition, serum cholesterol and triglycerides were elevated in the LPM (*Figure 1F*). These results confirm the iron accumulation with age post menopause (*Cheng et al., 2017*; *Jian et al., 2009*; *Milman et al., 2000*; *Milman et al., 1992*) and suggest a negative correlation between iron and ERα levels.

### AS aggravates in E₂-treated LPM *Apoe⁻/⁻* mice with reduced ERα expression and accumulation of body iron

To determine whether the effect of HRT was atheroprotective in postmenopausal females, OVX was performed to mimic postmenopause in *Apoe⁻/⁻* female mice at 8 wk of age (*Figure 2A*). The mice started to be fed high-fat chow 1 wk post OVX. Ages of 17 wk (9 wk after OVX) and 29 wk (21 wk after OVX) were considered as EPM and LPM stages, respectively. Peritoneal E₂ injection was performed and the serum E₂ was measured at the end points (*Figure 2—figure supplement 1A*). Although E₂ administration in the EPM significantly reduced plaque formation and mouse weight, it remarkably promoted atherosclerotic development and weight gain in late application (*Figure 2B and C*, *Figure 2—figure supplement 1B*), exactly as observed in humans (*Hlatky et al., 2002*; *Rossouw*

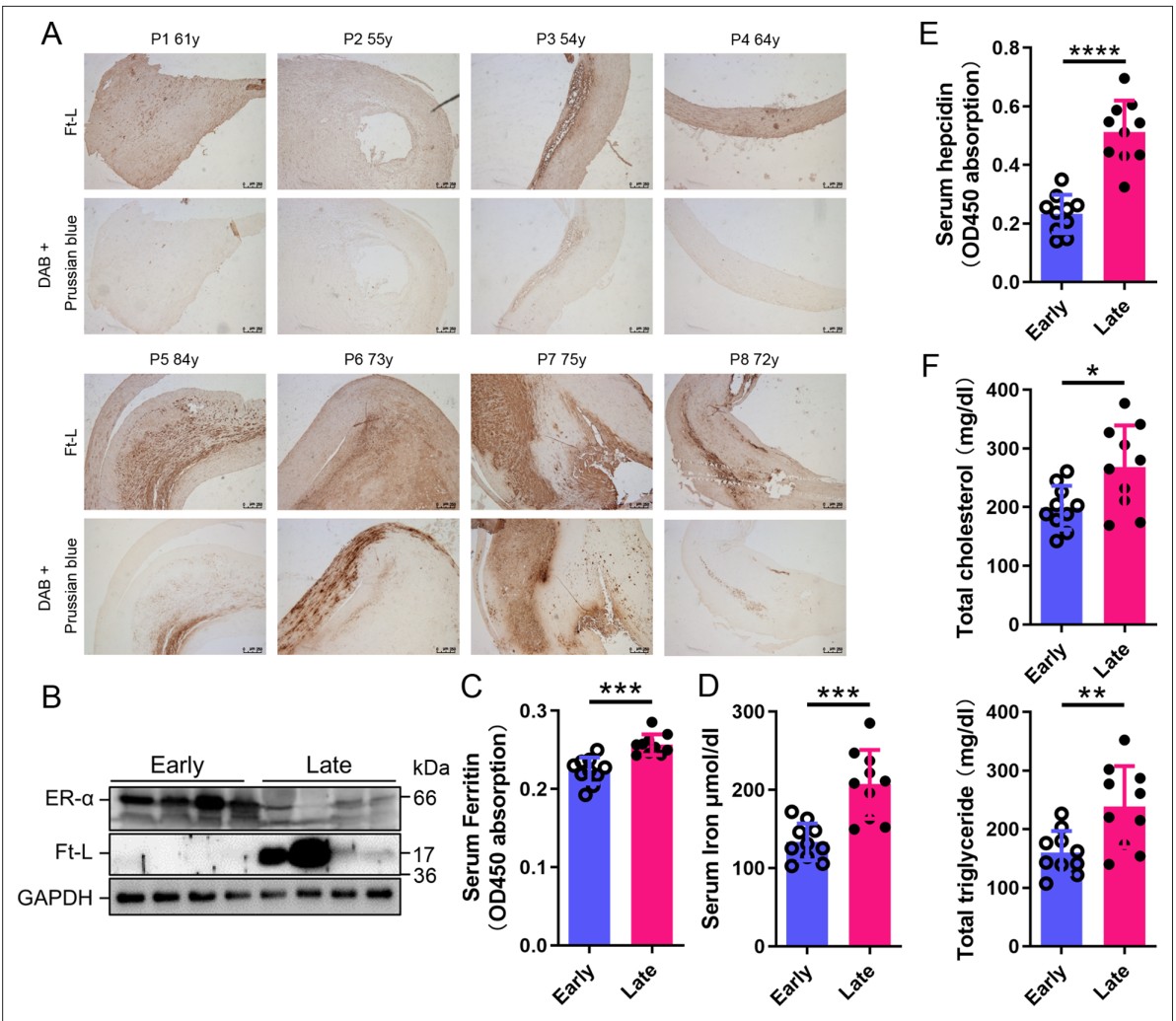

**Figure 1.** Estrogen receptor α (ERα) levels were negatively associated with iron content in human plaques. (**A**) Ferritin (Ft-L), revealed by immunohistochemistry (IHC), and iron content, revealed by DAB-enhanced Prussian blue staining, in plaque paraffin sections of eight postmenopausal patients. Upper panel: the early postmenopausal (EPM) group (P1–P4,<65 years old); lower panel: the late postmenopausal (LPM) group (P5–P8, >65 years old). (**B**) ERα and Ft-L expression in plaques measured by western blotting. The samples are the same as in (**A**) except for the lane switch between lane 6 (P6) and lane 7 (P7) by chance. (**C**) Serum ferritin levels in EPM (blue) and LPM (magenta) patients, detected by ELISA. n = 10/group, ***p<0.001. (**D**) Serum iron measured by using an autochemical analyzer (Beckman Coulter AU5421, CA). n = 10/group, ***p<0.001. (**E**) Serum hepcidin levels detected by ELISA. n = 10/group, ****p<0.0001. (**F**) Serum total cholesterol (left) and total triglyceride (right) levels. n = 10/group, *p<0.05, **p<0.01. Student's *t*-test analysis was used for (**C–F**).

The online version of this article includes the following source data for figure 1:

**Source data 1.** Raw data for *Figure 1*.

*et al., 2002*). We then detected aortic ERα expression to evaluate whether ERα was responsive to $E_2$ treatment. The results showed markedly lower ERα protein levels in the LPM mice than in the EPM mice (*Figure 2D*). More strikingly, ERα expression was further reduced in LPM mice after $E_2$ treatment but remained constantly high in EPM mice (*Figure 2D*). It has been reported that ERα protects against atherosclerosis by promoting lipid efflux and endothelial homeostasis (*Wang et al., 2021*; *Zhao et al., 2021*). Hence, we assessed three ERα downstream proteins, ABCA1, a lipid exporter whose gene promoter is predicted to have ERE, VEGF, an activator of angiogenesis, and eNOS, a modulator of vasoconstriction and vascular repair. They were all positively correlated with ERα expression (*Figure 2D*). Macrophage-derived foam cell formation is crucial in the development of atherogenesis (*Xu et al., 2021*). We, therefore, isolated peritoneal macrophages from early and late OVX mice after E2 treatment and found that ABCA1 expression responded to $E_2$ treatment similarly as observed in

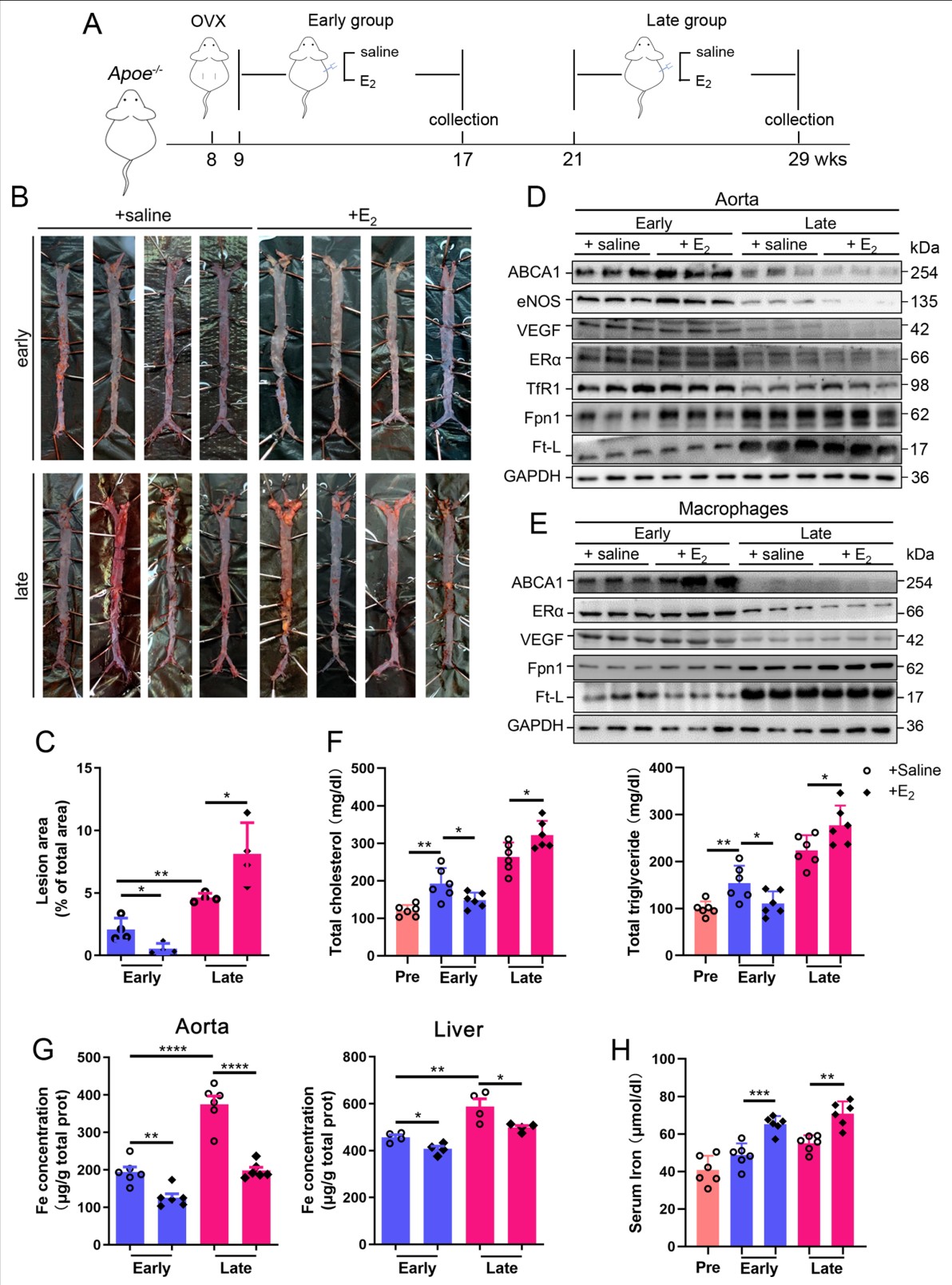

**Figure 2.** Atherosclerosis was aggravated in $E_2$-treated late postmenopausal $Apoe^{-/-}$ mice with lower ERα expression. (**A**) Flow diagram of mouse modeling. Early $E_2$-treatment group: ovariectomy (OVX) at 8 weeks old, 1-wk recovery, $E_2$ treatment for 8 wk; late $E_2$-treatment group: OVX at 8 weeks old, $E_2$ treatment from 21 weeks old to 29 weeks old for 8 wk. Saline is vehicle control. Mice were fed high-fat chow from 9 weeks old. (**B**) Oil red O-stained aortic lesions in $Apoe^{-/-}$ mice after $E_2$ treatment for 8 wk in the early postmenopausal (EPM) or late postmenopausal (LPM) group. (**C**) Statistical

*Figure 2 continued on next page*

*Figure 2 continued*

analysis of the area of atherosclerotic plaque in the aorta. n = 4/group, *p<0.05, **p<0.01. (**D**) The expression of iron-related or ERα-targeted proteins in the aorta, detected by western blotting. (**E**) Protein expression in peritoneal macrophages detected by western blotting. Macrophages were isolated from four mouse groups (early/late ± $E_2$, for details see 'Materials and methods'). (**F**) Serum total cholesterol and total triglyceride levels in the four mouse groups. Pre: serum samples before OVX as a control group. n = 6/group, *p<0.05, **p<0.01. (**G**) Iron content in aorta and liver, detected by ferrozine assays. n = 6/group, ****p<0.0001, **p<0.01, *p<0.05. (**H**) Serum iron in different groups, detected by using an autochemical analyzer (Beckman Coulter AU5421). n = 6/group, ***p<0.001, **p<0.01. Student's *t*-test analysis was used for (**C, F–H**).

The online version of this article includes the following source data and figure supplement(s) for figure 2:

**Source data 1.** Raw data for *Figure 2B–H*.

**Figure supplement 1.** Serum $E_2$ and body weight of postmenopausal *Apoe$^{-/-}$* mice.

**Figure supplement 1—source data 1.** Raw data of the mouse serum estradiol and body weight for *Figure 2—figure supplement 1*.

aortic tissue, with no change in ERα and VEGF, but keeping at high levels (*Figure 2E*). In line with this observation, serum cholesterol and triglycerides negatively correlated with ABCA1 (*Figure 2F*).

Our previous data have demonstrated that macrophage iron plays a critical role in the development of AS (*Cai et al., 2020*); therefore, iron-related proteins were monitored. In the isolated macrophages, ferritin was reduced in the EPM group, while Fpn1 was elevated in both stages in response to $E_2$ treatment (*Figure 2E*). These responses in tissue were very mild except Fpn1 in the early stage (*Figure 2D*), which suggests that not all cell types are responsive to $E_2$ in aorta tissue, particularly the disrupted iron homeostasis in the late stage. Iron levels in tissues, aorta and liver, were significantly higher in the LPM mice (*Figure 2G*, LPM vs. EPM without $E_2$ treatment). Interestingly, $E_2$ treatment elevated serum iron while lowering tissue iron in both EPM and LPM stages (*Figure 2G, H and E* treatment vs. saline), suggesting impaired iron homeostasis in the plaque area, particularly in macrophages (*Figure 2E*) and confirming that estrogen modulates iron homeostasis as previously suggested (*Yang et al., 2012*).

## $E_2$ downregulates ERα expression in an iron-dependent manner

Next, we aimed to identify whether aging or iron overload alone could trigger a decrease in ERα expression in vivo. To address this question, myeloid-specific *Fpn1* knockout mice (*Fpn1$^{Lyz2/Lyz2}$*) were used as a macrophage-iron overload model in the *Apoe$^{-/-}$* background (*Cai et al., 2020*). This double knockout (KO) model is considered relevant for AS studies due to the accumulation of a large number of macrophages in plaques, which contributes to the progression of atherosclerosis (*Moore et al., 2013*). The *Fpn1*-KO efficiency was determined (*Figure 3—figure supplement 1A*). OVX was performed in female mice fed standard chow at 16 wk or 40 wk of age, and $E_2$ was injected to model HRT, as illustrated in *Figure 3A*. *Figure 3B and C* show the severity of AS, which was significantly enhanced in the $E_2$-treated groups at both ages compared to the saline groups of *Apoe$^{-/-}$ Fpn1$^{Lyz2/Lyz2}$*. Notably, macrophage *Fpn1* KO mice displayed a larger lesion area after $E_2$ treatment at the EPM and LPM stages (*Figure 3C*), suggesting a dominant influence of iron on the effects of the $E_2$ treatment. In particular, the specific iron overload in macrophages, characteristic of the mouse model used, was sufficient to cause a significant increase in the lesion area in the LPM group (*Figure 3C*, lower panel), reproducing previous observations (*Cai et al., 2020*). We then examined the iron status in tissues and serum. Iron levels in tissues (aorta/liver) were higher in *Apoe$^{-/-}$ Fpn1$^{Lyz2/Lyz2}$* compared to *Apoe$^{-/-}$* mice, as revealed by ferrozine assays (*Figure 3—figure supplement 1B* for EPM groups and *Figure 3D* for LPM groups) and further supported by higher ferritin content (*Figure 3—figure supplement 1B* for EPM and *Figure 3E* for LPM). On the contrary, serum iron was lower in *Apoe$^{-/-}$Fpn1$^{Lyz2/Lyz2}$* mice at the EPM and LPM stages (*Figure 3F* and *Figure 3—figure supplement 1C*). However, $E_2$ administration did not significantly lower the iron and ferritin levels in the aorta and liver (*Figure 3D and E*, *Figure 3—figure supplement 1B*), confirming that Fpn1 acts as an iron exporter and that macrophages play a crucial role in response to $E_2$ treatment. $E_2$ administration significantly increased serum iron levels in the LPM group (*Figure 3F*) and mildly increased serum iron levels in the EPM group (*Figure 3—figure supplement 1C*), suggesting that other factors may contribute to the aging-related response to estrogen. Of note, ERα was downregulated in the aorta of the *Fpn1$^{Lyz2/Lyz2}$* mice and further downregulated by $E_2$ treatment, accompanied by reduced expression of ABCA1 and VEGF (*Figure 3E*, *Figure 3—figure supplement 1D*), suggesting that both iron alone and iron plus $E_2$ could

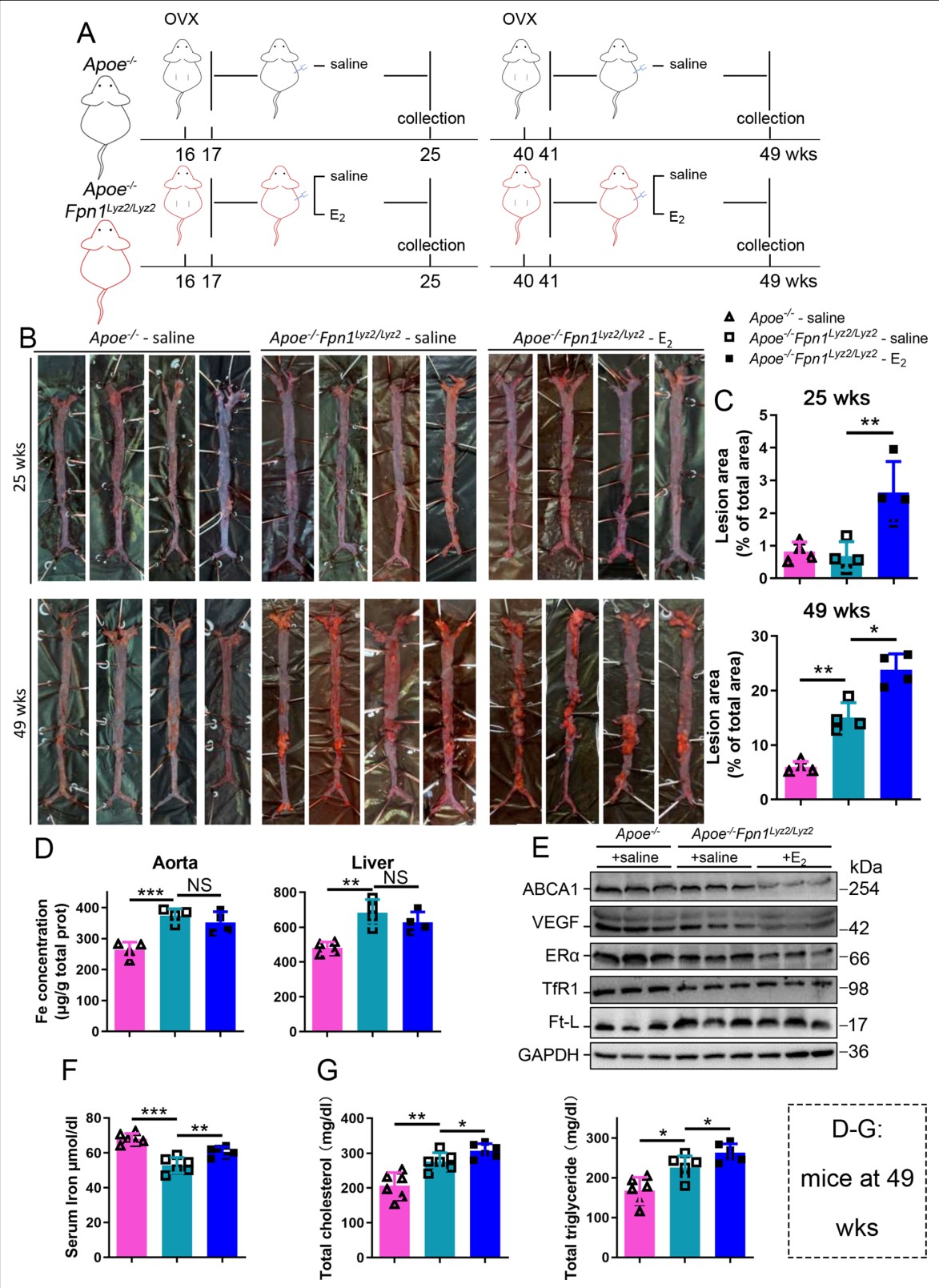

**Figure 3.** E₂-triggered estrogen receptor α (ERα) deficiency was observed in a genetic iron overload mouse model at postmenopausal age. (**A**) Flow diagram of mouse modeling. Early groups: ovariectomy (OVX) at 16 weeks old, 1-wk recovery, ± E₂ treatment for 8 wk; late groups: OVX at 40 weeks old, 1-wk recovery, ± E₂ treatment for 8 wk. Saline is vehicle control. The mice were fed with normal chow. (**B**) Oil red O-stained aortic lesions in *Apoe⁻/⁻* and *Apoe⁻/⁻ Fpn1^Lyz2/Lyz2* mice after E2 treatment for 8 wk in the early postmenopausal (EPM) or late postmenopausal (LPM) groups as indicated. (**C**) The lesion

*Figure 3 continued on next page*

*Figure 3 continued*

area in the aorta. n = 4/group, \*\*p<0.01, \*p<0.05. (**D**) The iron content of the aorta and liver detected by ferrozine assays. n = 6/group, \*\*\*p<0.001, \*\*p<0.01. (**E**) The expression of iron-related or ERα-targeted proteins in the aorta, detected by western blotting. (**F**) Serum iron level in different groups. n = 6/group, \*\*\*p<0.001, \*\*p<0.01. (**G**) Serum total cholesterol and total triglyceride levels. n = 6/group, \*p<0.05, \*\*p<0.01. The samples for (**D–G**) were from 49-week-old *Apoe*$^{-/-}$ and *Apoe*$^{-/-}$ *Fpn1*$^{Lyz2/Lyz2}$ mice. Student's *t*-test analysis was used for (**C, D, F, G**).

The online version of this article includes the following source data and figure supplement(s) for figure 3:

**Source data 1.** Raw data for *Figure 3B–G*.

**Figure supplement 1.** E$_2$-triggered estrogen receptor α (ERα) deficiency was observed in *Apoe*$^{-/-}$ *Fpn1*$^{Lyz2/Lyz2}$ at early postmenopause (25 weeks old).

**Figure supplement 1—source data 1.** Raw data for *Figure 3—figure supplement 1*.

downregulate ERα expression in macrophages. Consistent with the severity of atherosclerosis and the alteration of ERα and its target gene ABCA1, serum cholesterol and triglycerides were increased in the *Fpn1*$^{Lyz2/Lyz2}$ mice and further increased by E$_2$ treatment (*Figure 3G*).

## E$_2$ treatment potentiates iron-induced downregulation of ERα in both macrophages and endothelial cells

To further validate the interaction between iron and E$_2$ on ERα downregulation in different cell types, we used the macrophage-like cell line J774a.1, primary peritoneal macrophages from C57BL/6 female mice, and human umbilical vein endothelial cells (HUVECs). The cells were treated with E$_2$, ferric ammonium citrate (FAC, an iron source), and/or deferiprone (DFP, an iron chelator). Downregulation of ERα expression triggered by FAC with or without E$_2$ was observed in time- and concentration-dependent manners (*Figure 4A, B*). Such downregulation was confirmed in all tested cell types and could be partially suppressed by iron chelation (*Figure 4C–F*).

To examine the capacity of lipid export when ERα was downregulated, we loaded J774a.1 with oxidized low-density lipoprotein (oxLDL) and observed significantly more lipid accumulation in the E$_2$-treated plus iron overload group than in the other groups (*Figure 4G*), suggesting a tendency of macrophages to be converted into foam cells. Angiogenesis assays were also performed and showed that E$_2$, together with iron, inhibited angiogenesis (*Figure 4H*), which has been demonstrated to increase the risk of macrophage adhesion and intraplaque hemorrhage (*Chang and Nguyen, 2021*; *Mao et al., 2020*). The reduced levels of eNOS were also revealed by ELISA in HUVECs treated with E$_2$ and iron (*Figure 4I*). Overall, our data strongly support that both macrophages and endothelial cells are the effectors of E$_2$ in iron-mediated worsening by downregulation of ERα in the development of AS.

## Proteasome-mediated ERα degradation results from the interactive effects of iron overload and E$_2$ treatment mediated by the E3 ligase Mdm2

We wondered how excess iron and E$_2$ together downregulated ERα expression. To elucidate the underlying mechanism, we first searched for what regulates ERα expression. It was reported that the downregulation of ERα could be attributed to the methylation of its promoter region, which could be induced by oxidative stress (*Lung et al., 2020*). Because iron overload has been strongly correlated with oxidative stress, two relevant antioxidative enzymes, catalase (CAT) and superoxide dismutase (SOD) 2, were evaluated. The results showed no significant difference between the control and E$_2$/FAC treatments (*Figure 5—figure supplement 1A, B*). In addition, the mRNA level of ERα was examined and barely showed significant changes after FAC and E$_2$ treatment (*Figure 5—figure supplement 1C*), suggesting that post-transcriptional regulation of ERα better explained the decreased presence of the receptor after iron and/or E$_2$ treatment.

As stated, ERα may be regulated through an estrogen-ERα binding-dependent ubiquitination signaling pathway for degradation and estrogen recycling. To test this possibility, we treated J774a.1 cells with MG132, a proteasome inhibitor, and observed that ERα protein levels were significantly elevated in the presence of E$_2$ and excess iron (*Figure 5A*). Treatment with cycloheximide (CHX), an inhibitor of eukaryotic translation, showed that the half-life of ERα was shortened in the E$_2$ + FAC group (*Figure 5B*), suggesting a faster turnover rate of ERα in the presence of E$_2$ plus excess iron, verifying the activation of ERα proteasome degradation pathway (*Zhou and Slingerland, 2014*). We

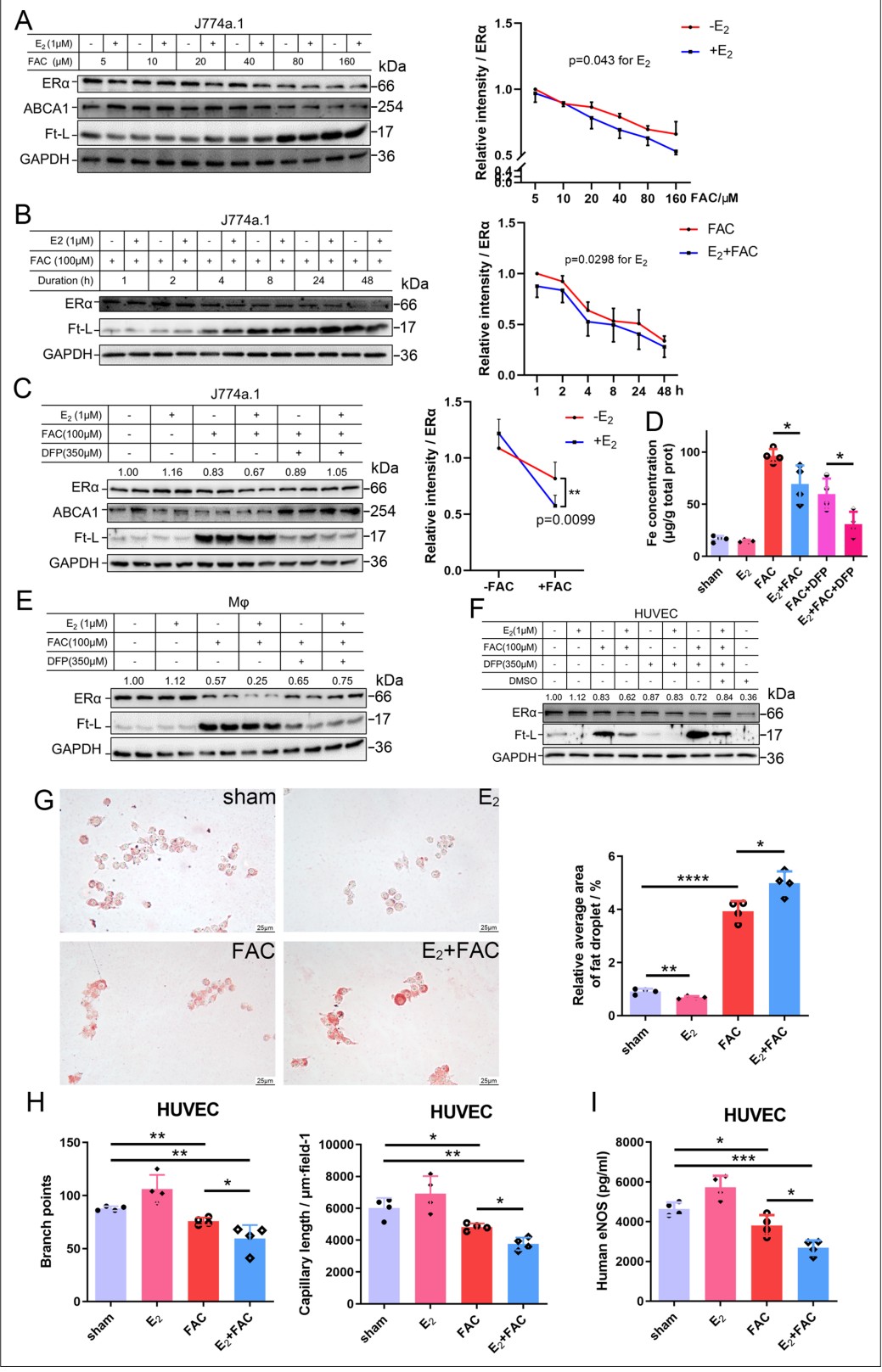

**Figure 4.** E₂ treatment potentiates iron-induced downregulation of estrogen receptor α (ERα) in vitro. (**A, B**) Left: ERα expression in the presence or absence of E₂ under different iron concentration conditions (**A**) or in the time course (**B**). Right: quantification data using ImageJ analysis. Two-way ANOVA was used. (**C**) Left: the rescue effect of iron chelation on the downregulation of ERα by FAC or FAC plus E₂. Right: quantification data using ImageJ

*Figure 4 continued on next page*

*Figure 4 continued*

analysis. Two-way ANOVA was used for the former four groups. (**D**) The intracellular iron content in J774a.1 under different iron-concentration conditions in the presence or absence of $E_2$, detected by ferrozine assays. n = 4, *p<0.05. (**E, F**) ERα expression in peritoneal macrophages (**E**) and human umbilical vein endothelial cells (HUVECs) (**F**) under the indicated iron and E2 conditions. (**A–C, E, F**) are data from western blotting. The quantification is indicated as a relative intensity of ERα (n = 4). (**G**) Oil red O-stained J774a.1 cells after treatment with FAC and/or $E_2$ (left) followed by oxidized LDL uptake, quantified by the area of droplets (right). scale bar = 25 μm, n = 4, ***p<0.001. (**H**) HUVEC angiogenesis assays, revealed by the number of branch points (left) and capillary length (right). n = 4, *p<0.05, **p<0.01. (**I**) eNOS level in HUVEC, assessed by ELISA. n = 4, *p<0.05, **p<0.01, ***p<0.001. Two-way ANOVA was used for (**A–C**). Student's *t*-test analysis was used for (**D, G–I**).

The online version of this article includes the following source data for figure 4:

**Source data 1.** Raw data for *Figure 4A–F and I*.

**Source data 2.** Raw data including parts of images and quantification for *Figure 4G*.

**Source data 3.** Raw data including the images for parts of *Figure 4G*.

**Source data 4.** Raw data including the images and quantification for *Figure 4H*.

then detected the ubiquitination levels of ERα by immunoprecipitation and immunoblotting. The results showed much more ubiquitinated and degraded ERα in the presence of $E_2$ and excess iron than in other conditions (*Figure 5C*), further supporting the proteolysis-dependent pathway for ERα degradation. We then tested a few E3 ligases (BRCA1, AHR, and Mdm2) in J774a.1, which were supposed to regulate ERα (*Fan et al., 2001*; *Khan et al., 2006*; *Saji et al., 2001*), and did not find a negative correlation between BRCA1/AHR and ERα in response to FAC or $E_2$ + FAC treatments (*Figure 5—figure supplement 2*). However, *Mdm2* was upregulated in the FAC and $E_2$ + FAC groups (*Figure 5D*). Treatment of the J774a.1 cells and HUVECs with the Mdm2 inhibitor Nutlin-3 demonstrated iron-dependent Mdm2-mediated degradation of ERα (*Figure 5E and F*). Importantly and consistently, Mdm2 expression was upregulated at the LPM stage compared with the EPM stage both in female mice's aorta and in AS patients' plaques (*Figure 5G and H*), which is precisely opposite to ERα expression (*Figure 1B*), and this effect was significantly enhanced at the LPM stage mice when E2 was administrated (*Figures 2D and E and 5G*). Our results indicate that Mdm2 is responsible for $E_2$-triggered ERα deficiency under iron overload conditions or in LPM women.

## Iron restriction therapy restores ERα levels and attenuates E2-triggered progressive atherosclerosis in late postmenopausal mice

To further verify whether iron overload is responsible for the $E_2$-induced downregulation of ERα and progressive atherosclerosis in LPM mice, we evaluated the effects of iron restriction. 21-week-old female *Apoe*[-/-] mice, OVX-ed at 8 wk of age, received iron chelation therapy through peritoneal injection of DFP (80 mg/kg) daily for 8 wk (*Figure 6A*). Similar to the definition used in *Figure 2*, 13 wk after OVX was considered as the LPM stage. Indeed, iron chelation attenuated the plaque-accelerated development of AS (*Figure 6B, C*). The contents of serum cholesterol and triglycerides compared with those in the $E_2$-only group were significantly diminished (*Figure 6D*). Consistent with previous data (*Figures 2 and 3*), ferrozine assays proved decreased iron deposition in tissues but increased iron in serum post $E_2$ application (*Figure 6E, F*). Although DFP administration reduced the tissue and serum iron levels, it did not induce anemia (*Figure 6F*, *Figure 6—figure supplement 1*), which was comparable to the mice at EPM (*Figure 2H*). We then detected aortic ERα expression and found significant upregulation by iron restriction, along with the upregulation of ABCA1 and VEGF (*Figure 6G*). In contrast, iron chelation by DFP significantly reduced Mdm2 expression (*Figure 6G*). In agreement with our previous findings in cell-based assays, these results corroborate the concept that late postmenopausal HRT-induced ERα deficiency is, at least partially, iron overload-mediated. Thus, the non-atheroprotective effects of $E_2$ in the LPM result from aging-mediated iron accumulation.

## Discussion

Estrogen has long been considered atheroprotective and responsible for the low morbidity of cardiovascular diseases in premenopausal women (*Lobo, 2017*; *Moss et al., 2019*). However, epidemiological

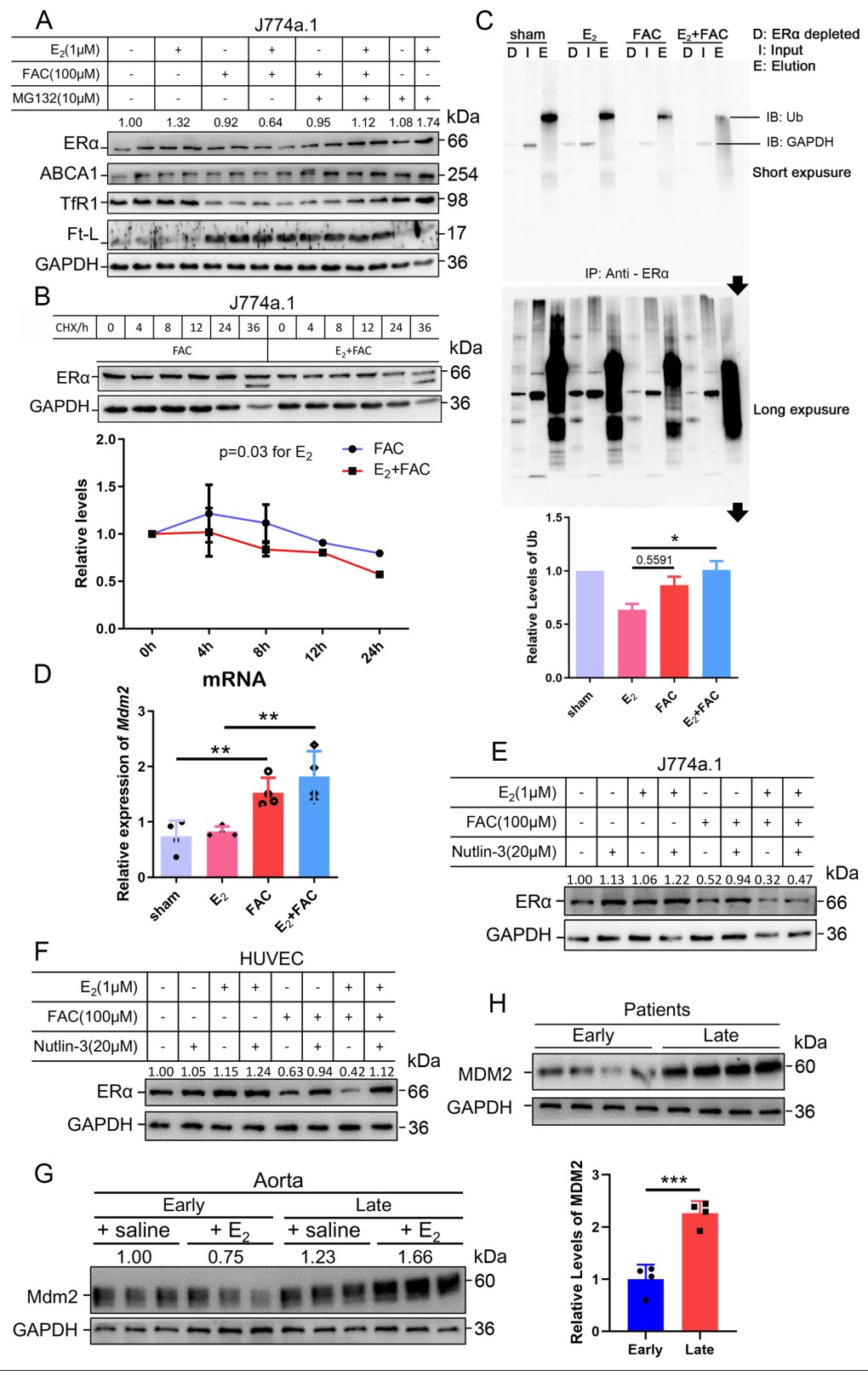

**Figure 5.** The interactive effects of iron overload and $E_2$ treatment on estrogen receptor α (ERα) downregulation are mediated by the E3 ligase MDM2. (**A**) Evaluation of ERα proteasome-dependent degradation in J774a.1 cells by western blotting. MG132: 10 μM. n = 4. (**B**) ERα turnover rate in J774a.1 cells under FAC or $E_2$ + FAC conditions, detected by western blotting after 20 μM cycloheximide (CHX) treatment. *p<0.05 using two-way ANOVA.

*Figure 5 continued on next page*

*Figure 5 continued*

(**C**) Ubiquitination of ERα, evaluated by western blotting (anti-ubiquitin) following immunoprecipitation against ERα antibody. n = 3, *p<0.05. (**D**) Relative *Mdm2* mRNA expression in J774a.1 cells, assessed by qPCR, n = 4, **p<0.01. (**E**) The protein levels of ERα in the presence of FAC or FAC plus $E_2$ in J774a.1 cells after treatment of Nutlin-3, a specific antagonist of Mdm2. n = 3. (**F**) The protein levels of ERα in the presence of FAC or FAC plus $E_2$ in human umbilical vein endothelial cells (HUVECs) after treatment of Nutlin-3. n = 3. (**G**) Mdm2 protein expression in the aortas of mice in the early postmenopausal (EPM) or late postmenopausal (LPM) stage, as detected by western blotting. n = 3/group. (**H**) MDM2 protein levels in patient plaques, detected by western blotting and quantified with ImageJ. n = 4/group, ***p<0.001. Two-way ANOVA was used for (**B**). Student's *t*-test analysis was used for (**C, D H**).

The online version of this article includes the following source data and figure supplement(s) for figure 5:

**Source data 1.** Raw data for *Figure 5*.

**Figure supplement 1.** No significant oxidative-stress was raised by application of $E_2$ and iron within the indicated concentration.

**Figure supplement 1—source data 1.** Raw data for *Figure 5—figure supplement 1*.

**Figure supplement 2.** E3-ligase responses to iron and $E_2$ treatment in different cell types.

**Figure supplement 2—source data 1.** RAW data for *Figure 5—figure supplement 2*.

studies of the Women's Health Initiative questioned the beneficial effects of late postmenopausal HRT (*Hlatky et al., 2002*). One hypothesis to explain the lack of beneficial effects of late postmenopausal HRT is that iron potentiates the adverse effects of estrogen in AS (*Sullivan, 1981*; *Sullivan, 2003*); however, a comprehensive in vivo study to test this hypothesis was missing. We reported previously that the developmental course of AS was highly accelerated in *Apoe^-/-Fpn1^Lyz2/Lyz2* mice compared with *Apoe^-/-* (*Cai et al., 2020*). The present study provides the first experimental evidence that iron overload facilitates ERα proteolysis, which is potentiated in the presence of $E_2$ and reverses the anti-atherogenic effect of $E_2$ (*Figure 7*). Our results support the benefit of early application of estrogen postmenopause. We propose that the combination of HRT and iron restriction therapy might be a long-term strategy for the preventive effects of $E_2$ from the development of AS in postmenopausal women.

The controversy of whether or not to proceed with HRT in postmenopausal women is fueled by an increased, although small, risk of breast cancer and the potentially harmful effect on cardiovascular outcomes (*Lobo, 2017*). Previous randomized trials did not consider the ages sorting out EPM from LPM and have not excluded subjects with iron depletion or loss in the recruited postmenopausal subjects (*Sullivan, 2003*). We sorted the recruited female AS volunteers from the Department of Vascular Surgery of Nanjing Drum Tower Hospital as EPM and LPM groups to reveal whether aging-associated iron deposition correlates with ERα expression. The negative correlation between age-related systemic iron status and intraplaque ERα expression was proposed, which prompted us to address the role of iron in ERα expression. However, the inverse relationship between iron and ERα levels needs to be further studied.

Previous efforts focused more on the role of estrogen in iron metabolism (primarily the hepcidin/Fpn axis) rather than vice versa (*Hou et al., 2012*; *Ikeda et al., 2012*; *Yang et al., 2012*). Both genes encoding hepcidin and Fpn are inhibited by $E_2$ treatment through an ERE binding (*Hou et al., 2012*; *Qian et al., 2015*; *Yang et al., 2012*). However, it was also reported that hepcidin expression decreased in the livers of OVX mice through a GPR30-BMP6-dependent mechanism, independent of the ERE-mediated $E_2$-ERα pathway (*Ikeda et al., 2012*). Despite the difference in hepcidin expression in OVX mice (*Bowling et al., 2014*; *Gavin et al., 2009*; *Hou et al., 2012*; *Ikeda et al., 2012*), the rest of the observations consistent with the present study are that aging, OVX, and genetic manipulation of Fpn induced progressive iron retention in tissues, accompanied by reduced ERα expression. $E_2$ administration further enhanced this reduction in ERα levels under the conditions mentioned above. Overall, high ERα levels are found in reproductive women, despite fluctuations caused by the periodic estrogen wave and blood loss in reproductive women (*Gavin et al., 2009*). The aging process, particularly in late postmenopausal women, progressively elevates iron levels and downregulates ERα, resulting in insufficient ERα to respond to $E_2$ treatment. $E_2$ treatment makes LPM mice gain weight with increased serum total cholesterol and triglyceride. Therefore, HRT is unlikely to result in an effective outcome in LPM women as in EPM women unless coupled with an iron-chelating scheme. This

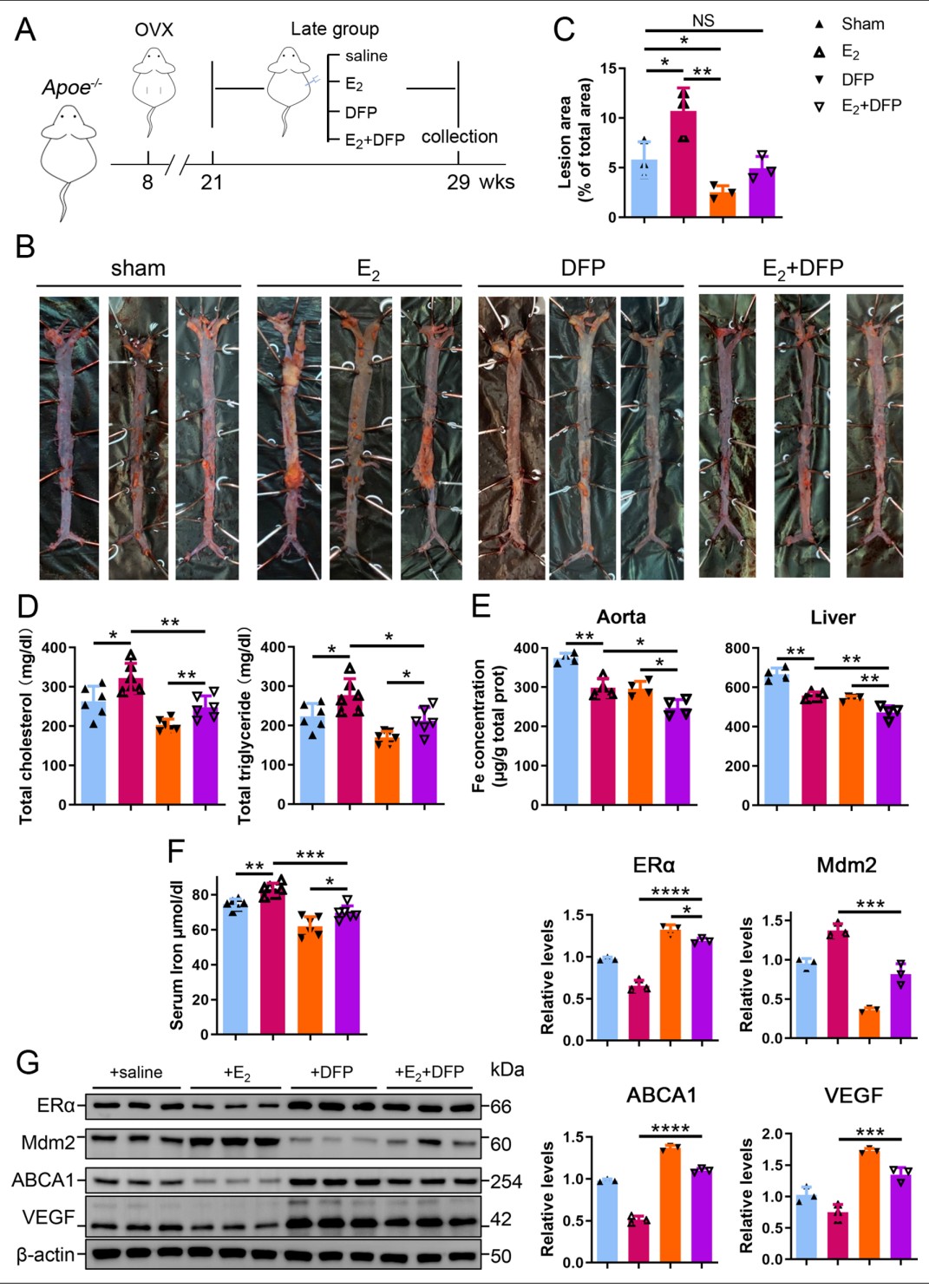

**Figure 6.** Iron restriction therapy restored estrogen receptor α (ERα) levels and attenuated $E_2$-triggered progressive atherosclerosis in late postmenopausal mice. (**A**) Flow diagram of mouse modeling. The mice were ovariectomized (OVX) at 8 weeks old and $E_2$, DFP or $E_2$ + DFP treated from 21 weeks old to 29 weeks old for 8 wk. Saline is vehicle control. Mice were fed high-fat chow one week after OVX. 13 wk post-OVX is considered as late postmenopause. (**B**) Oil red O-stained aortic lesions in *Apoe*$^{-/-}$ mice treated with $E_2$, DFP or $E_2$ + DFP as indicated. (**C**) The quantified lesion area of atherosclerotic plaques in the aorta from (**B**). n = 3, **p<0.01, *p<0.05. (**D**) Serum total cholesterol and total triglyceride levels. n = 6, *p<0.05, **p<0.01. (**E**) The iron content in the aorta and liver, detected by ferrozine assays. n = 4, **p<0.01, *p<0.05. (**F**) Determination of serum iron in different groups. n

*Figure 6 continued on next page*

*Figure 6 continued*

= 6, ***p<0.001, **p<0.01, *p<0.05. (**G**) Protein expression in the aorta, detected by western blotting (left) and quantified with ImageJ (right). n = 3. ****p<0.0001, ***p<0.001, *p<0.05. Student's *t*-test analysis was used for (**C–G**).

The online version of this article includes the following source data and figure supplement(s) for figure 6:

**Source data 1.** Raw data for *Figure 6B–G*.

**Figure supplement 1.** Serum hemoglobin after DFP administration with Student's *t*-test analysis (80 mg/kg, daily for 8 wk, *Apoe*$^{-/-}$, age 8 wk for Control, and 29 wk for the late, n = 4).

**Figure supplement 1—source data 1.** Hemoglobin values for *Figure 6—figure supplement 1*.

failure is because aggravated AS in LPM women is, at least partially, the result of age-related iron accumulation. We demonstrated the effectiveness of iron chelation in improving HRT outcomes in the mouse model, but further work is required to translate this finding for clinical practice.

ERα is the main effector of estrogen on cardiovascular function (*Aryan et al., 2020*; *Meng et al., 2021*). Though the negative correlation between ERα and iron levels needs to be substantiated in a larger cohort, we wondered how iron downregulated ERα. Since several E3 ubiquitin ligases (i.e., CHIP, E6AP, BRCA1, BARD1, SKP2, and Mdm2) have been found to catalyze the covalent binding of ubiquitin to lysine residues of ERα (reviewed in *Tecalco-Cruz and Ramírez-Jarquín, 2017*), we tested and found that Mdm2 is responsive to iron treatment in cells and mice and negatively correlated with ERα, particularly under high iron conditions. Furthermore, we showed that Mdm2 is a negative regulator of ERα.

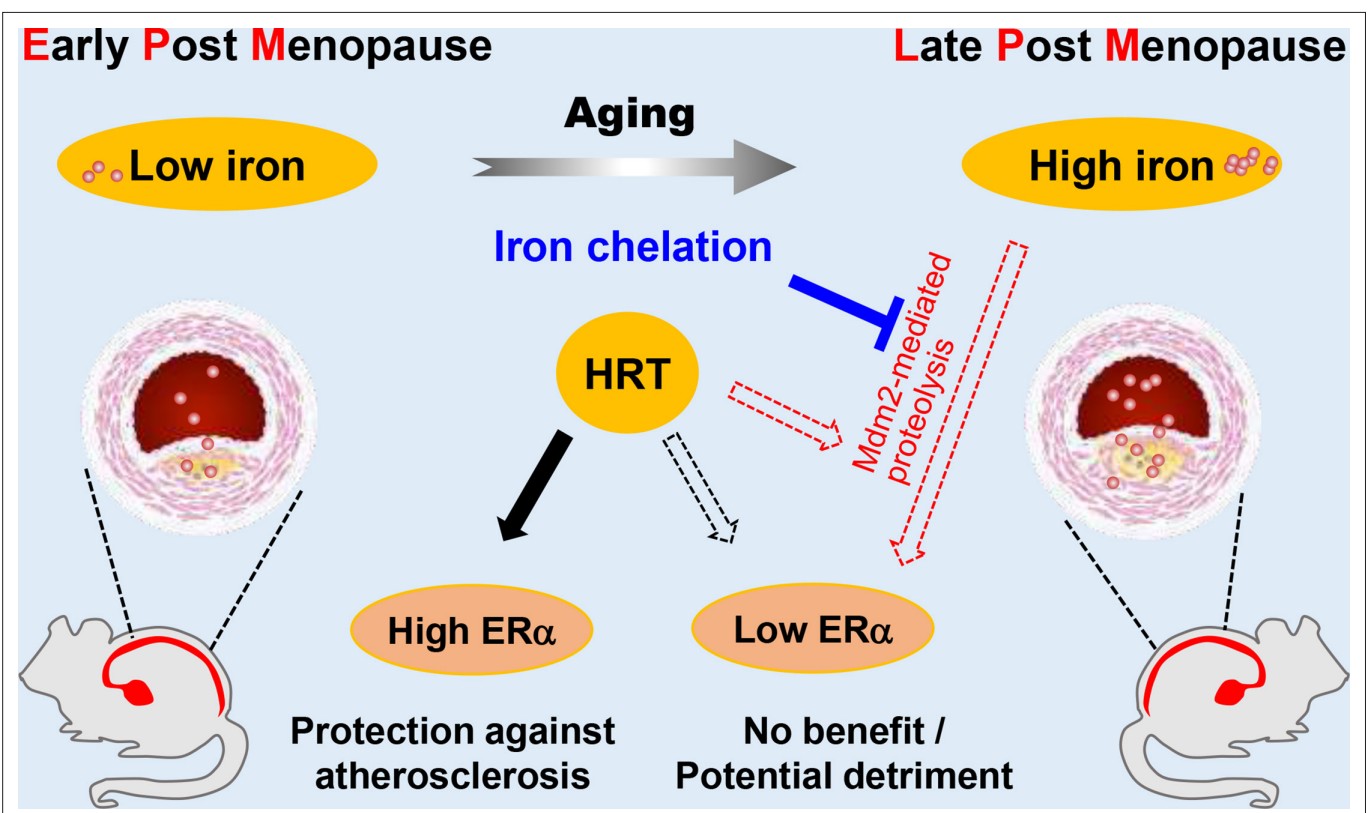

**Figure 7.** Schematic model for the effects of postmenopausal iron accumulation with or without hormone replacement therapy (HRT) on atherosclerosis (AS) severity through modulating estrogen receptor α (ERα) expression. Iron accumulation occurs naturally and gradually after menopause. In early postmenopausal (EPM), iron retention was mild, and ERα was responsive to HRT application to achieve protective effects. However, when iron overload is significant in late postmenopausal (LPM), Mdm2 is upregulated along with ERα downregulation. This negative correlation is potentiated by the application of HRT and iron accumulation with aging. Therefore, HRT use avails to aggravate the progression of AS in the LPM period. Iron chelation, however, reverses the adverse effect of HRT and attenuates the accelerated development of AS, suggesting a protective role of appropriate iron restriction in the LPM stage.

Our findings may be context specific, as some differences are noted in other cancer cell types (*Dongiovanni et al., 2010*; *Zhang et al., 2020*). Mdm2 acts as a ubiquitin ligase E3 to p53 in SV40 hepatocytes (*Honda et al., 1997*) and has been shown to act as a direct coactivator of ERα function in ERα-positive breast cancer (*Saji et al., 2001*). Nevertheless, iron-dependent downregulation was revealed in leukemia cell lines and primary human cells derived from acute myeloid leukemia patients (*Calabrese et al., 2020*), suggesting a cell-type-specific regulation of Mdm2 by iron. In our study, the upregulation of Mdm2 by $E_2$ occurred in the context of iron overload both in vivo and in vitro, concluding that Mdm2 is the critical mediator that participates in iron overload triggered ERα loss. One limitation in this study is that we have not figured out the mechanism by which aging and the combination of iron and $E_2$ favor Mdm2 expression.

In summary, this study demonstrates the impact of iron overload in $E_2$-mediated ERα proteolysis and its critical consequence on the outcome of HRT in the mouse model. With the efficacy of HRT challenged by 'the window of opportunity' theory (*Yesufu et al., 2007*), it would be vital to explore therapies that maintain ERα expression mediating the protective effects of estrogen. This study suggests that immediate HRT after menopause along with appropriate iron chelation might provide benefits from atherosclerosis. Our findings might also be applicable to other age and ERα-related postmenopausal diseases, such as osteoporosis. Additional studies are needed to validate these findings.

## Materials and methods
### Participants
Participants in this study included 20 postmenopausal (at least 1 y since menopause, without HRT) AS patients aged 54–84 y, recruited from the Vascular Surgery Department, The Affiliated Drum Tower Hospital, Nanjing University Medical School (*Table 1*). Patients were divided into early (55–65 years old) and late (>65 years old) groups since menopause for over 10 y was defined as late postmenopause. 20 fasting serum samples were collected at the outpatient service. Of them, eight patients undergoing carotid endarterectomy were recruited, and plaque samples were collected immediately after separation. All patients provided written informed consent. Exclusion criteria included the current use of oral contraceptives or other medications. Further details on the exclusion criteria were referenced (*Wactawski-Wende et al., 2009*). This study complies with the Declaration of Helsinki, and the Institutional Review Board of Nanjing Drum Tower Hospital, the Affiliated Hospital of Nanjing University Medical School, approved the study.

### Animals
*Apoe$^{-/-}$* mice were obtained from the Model Animal Research Center of Nanjing University (Nanjing, China). *Apoe$^{-/-}$ Fpn1$^{Lyz2/Lyz2}$* mice on the C57BL/6J background were generated in our previous study. For the control experiments in 'E2-treated LPM Apoe-/- mice with reduced ERα expression and accumulation of body iron' and 'Iron restriction therapy restores ERα levels and attenuates E2-triggered progressive atherosclerosis in late postmenopausal mice', female *Apoe$^{-/-}$* mice at the age of 8 wk (defined as premenopausal) were anesthetized and bilaterally ovariectomized through a 1 cm dorsal incision. After surgery, mice were allowed to recover for 1 wk and randomly divided into early and late groups. For each group, mice were fed a high-fat diet (0.2% cholesterol and 20% fat) and injected with saline, $E_2$ (3 μg/kg every other day, Solarbio, Beijing, China), or $E_2$ + DFP (80 mg/kg daily, Sigma–Aldrich, St. Louis, MO) for 8 wk from week 9 as the early OVX group or week 21 as the late OVX group.

For the control experiments included in Result 3, female *Apoe$^{-/-}$* and *Apoe$^{-/-}$ Fpn1$^{Lyz2/Lyz2}$* mice were ovariectomized at the age of 16 or 40 wk. $E_2$ injection (3 μg/kg every other day) was performed 1 wk after surgery for 8 wk. All animals were housed and fed standard chow in the SPF animal facility with an average 12 hr light-and-dark cycle and under controlled temperature conditions (25°C). The mice were anesthetized with an intraperitoneal injection of pentobarbital sodium (40 mg/kg) and euthanized by cervical dislocation for sample collection. The protocols were approved by the Animal Investigation Ethics Committee of The Affiliated Drum Tower Hospital of Nanjing University Medical School and were performed according to the Guidelines for the Care and Use of Laboratory Animals published by the National Institutes of Health, USA.

## Cell culture

J774a.1 cells, HUVECs, and MCF-7 cells were purchased from Cellcook (Guangzhou, China) with STR profiling report and cultured in DMEM (Gibco, Life Technologies, UK) supplemented with 10% fetal bovine serum (FBS). No mycoplasma contamination was found during experiment. Peritoneal macrophages were collected from peritoneal exudates 3 d after injecting 8-week-old mice with 0.3 ml of 4% BBL thioglycollate, Brewer modified (BD Biosciences, Shanghai, China), and then cultured in RPMI 1640 medium supplemented with 10% (FBS) for 8 hr. Macrophages were cultured in a medium containing 50 µg/ml human oxidized low-density lipoprotein (oxLDL) in the presence of 1 µM $E_2$, 100 µM ferric ammonium citrate (FAC), 350 µM DFP, or indicated combination for 48 hr as needed.

Oil Red O staining was performed to evaluate foam cell formation. The quantification was carried out with the following formula: relative average area of the fat droplet (%) = Target (Area of fat droplets/Numbers of cells)/Control (Area of fat droplets/Number of cells) * 100%. The area was analyzed using ImageJ software. Angiogenesis assays were performed to evaluate angiogenic capacity. Cellular iron levels were estimated using ferrozine assays. The protein levels of ERα, ABCA1, VEGF, TfR1, and Ft-L were determined by western blot analysis.

## Isolation of peritoneal macrophages from mice

The mice were intraperitoneally injected with 4% starch broth (NaCl 0.5 g, beef extract 0.3 g, peptone 1.0 g, and starch 3.0 g in 100 ml of distilled $H_2O$) 3 d before euthanasia. After anesthesia, the abdominal skin was carefully cut to 1 cm, and 5–8 ml PBS with 3% FBS was injected into the enterocoele. After 10 min of massage, the liquid was extracted and centrifuged ($1000 \times g$, 5 min). The sediment was then plated into 6-well plates for attachment or cryopreserved for further assays.

## Blood samples/tests and tissue collection

The mice were fasted for 1 d, then anesthetized with an intraperitoneal injection of pentobarbital sodium (40 mg/kg), and euthanized by cervical dislocation. Blood was drawn from the inferior vena cava and collected in heparinized tubes. Hemoglobin was measured by routine blood tests. Plasma was prepared by centrifugation ($1200 \times g$) for 15 min at 4°C and then stored at –80°C for the determination of serum iron, lipid estrogen, and cytokine levels. The mice were then perfused with 4°C saline through the left ventricle. After perfusion, the arteries, hearts, livers, and spleens were harvested. The samples were fixed in 4% paraformaldehyde or quickly frozen at –80°C for further analysis.

## Serum lipid content and lesion area in the aorta and the aortic root

Lipid content was determined with Oil Red O to stain the aorta. To assess the atherosclerotic lesion area, the aorta was analyzed from the aorta arch to the abdominal aortic bifurcation. The quantification of lesion area and size was performed using ImageJ software. Serum cholesterol and triglycerides were measured by the clinical laboratory of Nanjing Drum Tower Hospital using an autochemical analyzer (Beckman Coulter AU5421, CA).

## Immunohistochemistry (IHC) and Prussian blue staining

Sections of mouse aortic valve or patient carotid artery plaques were used to assess the plaque iron composition by IHC staining for Ft-L and Prussian blue staining with DAB enhancement for ferric iron. The primary antibody against Ft-L was made using recombinant human Ft-L subunit as antigen by GenScript (Nanjing, China).

Images were captured under a light microscope (Leica, Germany). For quantitative analysis of images, three sections per animal at intervals of 30 µm were analyzed. The intensity of positive staining was analyzed using ImageJ software.

## Iron assays

As previously described, deparaffinized tissue sections were stained with Prussian blue staining for nonheme iron (*Wang et al., 2016*). Serum iron was measured by the clinical laboratory of Nanjing Drum Tower Hospital using an autochemical analyzer (Beckman Coulter AU5421, CA). Total nonheme iron in the tissues was measured by colorimetric ferrozine-based assays as previously described (*Li et al., 2018*). Briefly, 22 µl concentrated HCl (11.6 M) was added to 100 µl of homogenized tissue samples (approximately 500 µg total protein). The sample was then heated at 95°C for 20 min, followed by

centrifugation at 12,000 × $g$ for 10 min. The supernatant was transferred into a clean tube. Ascorbate was added to reduce the $Fe^{3+}$ into $Fe^{2+}$. After 2 min of incubation at room temperature, ferrozine and saturated ammonium acetate ($NH_4Ac$) were sequentially added to each tube, and the absorbance was measured at 570 nm (BioTek ELx800, Shanghai, China) within 30 min.

## Determination of eNOS, serum hepcidin, and ferritin by ELISA

eNOS (Abcam, Cambridge, MA), serum hepcidin (Solarbio), and ferritin (US Biological, #F4015-11, Swampscott, MA) were detected by ELISA according to the manufacturer's protocols.

## Western blotting

Protein lysates were run in gels and transferred to membranes, as previously reported (*Cai et al., 2018*). The membranes were probed using antibodies directed against ERα, eNOS, and VEGF purchased from Servicebio (Wuhan, China), ABCA1, BRCA1, AHR, and SOD2 from Abcam, TfR1 from ProteinTech Group Inc (Chicago, IL), MDM2 from Abcam, GAPDH from Bioworld Tech. (St. Louis Park, MN), and ferritin L (Ft-L) made by using purified human ferritin L subunit as antigen by GenScript.

## Detection of catalase enzymatic activity

The catalase activities were measured following the manufacturer's protocols of the CAT assay kit (Jiancheng Bioengineering, Nanjing, China).

## Quantitative real-time PCR (qRT-PCR)

Total cellular RNA was isolated from peritoneal macrophages using TRIzol (Invitrogen, Carlsbad, CA) and reversely transcribed to cDNA. qRT–PCR experiments were performed with SYBR Green PCR master mixture (Thermo Fisher Scientific). The primer sequences were as follows: 5'-TTATGGGGTCTG GTCCTGTG-3' and 5'- CATCTCTCTGACGCTTGTGC-3' for *Esr1*, 5'-GCCACTGCCGCATCCTCTTC-3' and 5'- AGCCTCAGGGCATCGGAACC-3' for *Actin*, and 5'-TGTCTGTGTCTACCGAGGGTG-3' and 5'-TCCAACGGACTTTAACAACTTCA-3' for *Mdm2*.

## Immunoprecipitation (IP) and detection of ubiquitin

ERα proteins were immunoprecipitated from J774a.1 cell lysates according to the manufacturer's protocols (11204D, Invitrogen). The ERα antibody was the same as that used for western blotting and was purchased from Santa Cruz (1:200 dilution). Western blotting was used to detect the efficiency of IP and the level of ubiquitin.

## Angiogenesis assays

Matrigel (50 µl/well) was transferred to a 96-well plate, followed by inoculation of HUVECs ($2 \times 10^4$ cells) and treatment with the medicines described in the cell culture. After 8 hr, images were captured with an inverted microscope. The extent of tube formation was assessed by measuring branch points and capillary length using the 'Angiogenesis Analyser' plug-in designed by Gilles Carpentier with ImageJ software.

## Statistical analysis

All experiments were randomized and blinded. All experiments were technically and biologically replicated for at least three times. All the data are presented as the mean ± SD. A two-tailed Student's $t$-test (for two groups comparison) or one-way/two-way ANOVA followed by multiple-comparisons test with Bonferroni correction (for more than two groups comparison) was performed using SPSS 17.0 (SPSS Inc, Chicago, IL). $p < 0.05$ indicates statistical significance.

## Acknowledgements

We thank Dr. Deliang Zhang for giving valuable suggestions while writing the article. This work was supported by the National Natural Science Foundation of China [grant numbers: 32271221, 81870348].

## Additional information

### Funding

| Funder | Grant reference number | Author |
|---|---|---|
| National Natural Science Foundation of China | 32271221 | Kuanyu Li |
| National Natural Science Foundation of China | 81870348 | Tong Qiao |

The funders had no role in study design, data collection and interpretation, or the decision to submit the work for publication.

### Author contributions

Tianze Xu, Conceptualization, Data curation, Software, Validation, Visualization, Writing – original draft; Jing Cai, Resources, Supervision, Project administration; Lei Wang, Software, Validation, Investigation; Li Xu, Validation, Visualization, Methodology; Hongting Zhao, Formal analysis, Methodology; Fudi Wang, Resources; Esther G Meyron-Holtz, Fanis Missirlis, Writing - review and editing; Tong Qiao, Supervision, Funding acquisition, Project administration; Kuanyu Li, Conceptualization, Funding acquisition, Project administration, Writing - review and editing

### Author ORCIDs

Tianze Xu ⓘ https://orcid.org/0000-0002-5161-1245
Fanis Missirlis ⓘ http://orcid.org/0000-0003-0467-8444
Kuanyu Li ⓘ http://orcid.org/0000-0001-9738-049X

### Ethics

Human subjects: This study complies with the Declaration of Helsinki, and the Institutional Review Board of Nanjing Drum Tower Hospital, the Affiliated Hospital of Nanjing University Medical School, approved the study (serial number: 2021-368-02).

The protocols were approved by the Animal Investigation Ethics Committee of The Affiliated Drum Tower Hospital of Nanjing University Medical School (serial number: 2021-LCYJ-MS-18) and were performed according to the Guidelines for the Care and Use of Laboratory Animals published by the National Institutes of Health, USA.

### Decision letter and Author response

Decision letter https://doi.org/10.7554/eLife.80494.sa1
Author response https://doi.org/10.7554/eLife.80494.sa2

## Additional files

### Supplementary files

• MDAR checklist

### Data availability

All data generated or analysed during this study are included in the manuscript and supporting file. Source data files have been provided for all figures and figure supplements.

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
