## [Editor Report]

These important research findings provide new insights into the biology of aging and support an important role for iron accumulation in post-menopausal women as a major reason why estrogen therapy is not as effective in preventing atherosclerosis as it is in the pre-menopausal state. The evidence supporting the conclusions is compelling using both animals models and human tissues. This work will be of broad interest to researchers and clinicians.

---

## [Decision Letter]

**Decision letter after peer review:**

Thank you for submitting your article "Aging-related iron deposit prevents the benefits of HRT from late postmenopausal atherosclerosis" for consideration by *eLife*. Your article has been reviewed by 3 peer reviewers, one of whom is a member of our Board of Reviewing Editors, and the evaluation has been overseen by a Reviewing Editor and Carlos Isales as the Senior Editor. The reviewers have opted to remain anonymous.

Essential revisions:

1) The statement of higher serum iron in LPM versus EMP needs to be substantiated and the differences between serum iron, ferritin, and tissue iron need to be made more clear. Hepcidin levels would be of interest as well.

2) How do the authors explain the iron-dependent effect of E2 on increasing atherosclerosis in the macrophage FPN KO ApoE KO mice since E2 appears to restore tissue iron and serum iron back towards more normal levels.

3) Please check all statistical tests performed especially as they relate to reviewer #1 question 9.

4) Please check the comments of review 1 regarding Figure 6--DFP-only group (ie, without the E2)? Does DFP treatment of ApoE mice post OVX prevent the reduced ERalpha expression associated with LMP?

5) Clarification of human data shown in Figure 1 is needed and thorough check of confounders is needed.

*Reviewer #1 (Recommendations for the authors):*

1) In Figure 1B, can the authors explain why the LPM group has both higher serum iron and tissue iron? Can the authors point to a reference showing higher serum iron in LPM vs EPM? Measure of hepcidin levels would be of interest here as, higher hepcidin levels in the LPM group would explain the high tissue iron/ferritin, but would be expected to lead to lower serum iron levels.

2) In Figure 1E, Ferritin (Ft-L) levels are highly variable among the LPM group. The authors mentioned that the western blot samples in Figure 1E are the same as in figure 1A. However, Ft-L levels measured by IHC in sample P6 and P7 are similar but there are large differences in the Ft-L levels in sample P6 and P7 shown by western blot. Could the authors please explain the reason for this variability across the two techniques.

3) Line 288- Plaque ER expression was lower in LPM as compared to EPM. Please correct the words.

4) The correlation in Figure 1F is limited by sample size as well as batch effect (ie, the correlation does not exist within the LPM group or within the EPM group?) and should be removed.

5) In Figure 2E in the macrophages, a Western Blot of FPN1 should be included.

6) This sentence needs further clarification: "Ferritin was reduced in response to E2 treatment in the EPM stage (this appears to be only true in macrophages). In contrast, ferritin remained high in the late stage after E2 treatment in both aortae and isolated macrophages (Figure 2D and 2E), which could be explained by the response of Fpn1 expression (clarify this statement) that was decreased in the EPM stage but not in the LPM stage."

7) In Figure 3, how do the authors explain the iron-dependent effect of E2 on increasing atherosclerosis in the macrophage FPN KO ApoE KO mice since E2 appears to restore tissue iron and serum iron back towards more normal levels (ie, closer to the ApoE KO mice)? An ApoE + E2 control would also have been helpful here.

8) In Figure 3, the authors should provide a Western Blot for FPN showing the efficacy of the Cre-Lox model, particularly at the two ages.

9) The graphs depicted in Figures 4A and 4B have large error bars that intersect, making the conclusion that E2 and iron intersect to regulate ERalpha expression less convincing. Did the authors perform a 2-way ANOVA? What was the p-value for E2? What was the interaction p-value? Clearly, FAC alone decreases ERalpha expression substantially, so why do the authors think it is E2-dependent?

10) In Figure 5C, can the authors explain the relative levels compared to the sham group?

11) In Figure 5E, it is unclear why nutlin-3 does not prevent the E2+FAC-mediated reduction of ERalpha?

12) For the ApoE experiments, can the authors please provide trends in weight and the food consumptions for each of the mouse groups (for Figures 2, 3, and 6).

13) In Figure 6, do the authors have a DFP-only group (ie, without the E2)? Does DFP treatment of ApoE mice post OVX prevent the reduced ERalpha expression associated with LMP?

14) In Figure 2, treatment with E2 reduces atherosclerosis in ApoE mice in EPM. In Figure 6, the DFP is administered at the same time as E2. Would pre-treatment with DFP prolong an "EPM state" thus resulting in E2 causing a reduction in atherosclerotic lesions?

*Reviewer #2 (Recommendations for the authors):*

1) The human data as presented in Figure 1 is not convincing given multiple confounders may exist which might affect the data. List patient demographics and compare them directly (age, risk factors, symptomatic, chol lowering meds). What types of plaque were collected? I assume the rationale between late and early is that iron accumulates in late but not early? Make this rationale clear--it is important for the ENTIRE manuscript. Prussian blue is usually blue but I don't see any blue? Higher power please with more details.

2)Mice experiments. Needs to make clear why you looked at early versus late OVX? How was dose of E2 chosen and why. The micrographs of lipid staining are not particularly convincing. It looks as if there is no difference in the late group between saline and E2? Would be useful to show estrogen levels. Why does Fe drop in the late E2 and in Eary E2? Total body iron stores should increase in both groups according to me? Why not show ferritin levels in all mice?

3) In Figure 3 explain more clearly why these timepoints were chosen. You change timepoints from the previous figure why? What is the hypothesis being exposed? All of this is hard to follow.

4) IN Figure 4 why are endothelial cells introduced into the experiments?

5) In Figure 6 what does Dep do in the absence of OVX? This is a needed control. In addition the amount of Era increase with Dep is very small. Can this really explain the effects of Dep on atherosclerosis? Some verification via other methods is needed.

*Reviewer #3 (Recommendations for the authors):*

1. The clinical study involving 20 postmenopausal women is small. It would be important to state somewhere this observation of age-related inverse relationships between iron and ERa levels needs to be replicated in a larger cohort.

2. It is noted that the study subjects had known atherosclerosis. Do these findings bear out in postmenopausal in general? A comment on this matter seems appropriate.

3. Line 44-46 in Abstract: "HRT is recommended immediately after menopause along with appropriate iron chelation to protect from atherosclerosis" is too strong based on such small clinical sample and on single study. Please remove or restate.

4. Line 287-288 (page 10): "By contrast, plaque ERa expression was lower in the EPM than in LPM (Figure 1E)" is wrong. It should be opposite.

5. Line 355-358 (page 13): I do not understand what the authors mean by this.

6. Lines 396 (page 14): "massively correlated." Odd word usage. Suggest "strongly correlated"?

7. Line 421 (page 15): "Amazingly" implies the authors were surprised by the result. Suggest "Importantly…".

8. Scale back on clinical recommendations. Suggest "this study suggest XYZ." "Additional studies are need to validate these findings…", etc.

[Editors' note: further revisions were suggested prior to acceptance, as described below.]

Thank you for resubmitting your work entitled "Hormone replacement therapy for postmenopausal atherosclerosis is offset by late age iron deposition" for further consideration by *eLife*. Your revised article has been evaluated by Carlos Isales (Senior Editor) and a Reviewing Editor.

The manuscript has been improved but there are some remaining issues that need to be addressed, as outlined below:

*Reviewer #2 (Recommendations for the authors):*

The authors have done a good job of responding to my comments and the manuscript is improved and more easy to follow and read.

I few remaining comments:

Please tone down the abstract: "Thus, iron and estradiol together downregulate ERα through Mdm2-mediated proteolysis, explaining failures of HRT in late postmenopausal subjects with aging-related iron accumulation". The study provides one mechanism that may be important but does not explain the failure of HRT in late post postmenopausal subjects. This is a mouse study.

Similarly in the conclusions "This study suggests that immediate HRT after

379 menopause along with appropriate iron chelation would provide benefits from

380 atherosclerosis." Would implies confirmation--might is a better word.

The study demonstrates the role of iron overload in diminishing HRT benefits but it doesn't specifically show a certain mechanism. There certainly need to be some limitations mentioned in the discussion.

---

## [Author Response]

Essential revisions:1) The statement of higher serum iron in LPM versus EMP needs to be substantiated and the differences between serum iron, ferritin, and tissue iron need to be made more clear. Hepcidin levels would be of interest as well.

Thank you for this comment, which was addressed: “Tissue ferritin was evaluated by IHC (Figure 1A) and immunoblotting (Figure 1B), serum ferritin by ELISA (Figure 1C), tissue iron by DAB-enhanced Prussian blue staining (Figure 1A), and serum iron by ferrozine assays (Figure 1D). The results showed that ferritin and iron levels were significantly increased in plaques and serum of the LPM, compared to that in the EPM (Figure 1A-D)”. The serum hepcidin level was measured by ELISA, presented in Figure 1E.

2) How do the authors explain the iron-dependent effect of E2 on increasing atherosclerosis in the macrophage FPN KO ApoE KO mice since E2 appears to restore tissue iron and serum iron back towards more normal levels.

This is an interesting question. Our thoughts are as follows. Physiologically, E2 decreases tissue iron and increases serum iron and also increases iron uptake to compensate the periodic loss of iron in blood. Then, body iron gradually reach balance and estrogen level goes back to the bottom line. Cellular physiological response to estrogen starts with estrogen binding to either ERa or ERb. ERa is reduced in the macrophage FPN KO Apoe KO mice and administration of E2 may not reach the protective effects through estrogen receptors. However, it has been demonstrated that no protein-bound estrogen has the property to diffuse into cells freely with no regulation and estrogen may have pro- and anti-inflammatory properties depending on the circumstances including different tissues or diseases (PMID: 17640948). Plaques in macrophage FPN KO Apoe KO mice are supposed to challenge more inflammatory situation than the plaques in Apoe KO mice. Therefore, E2 exhibits possible paradox consequences due to timing of E2 administration and iron status.

3) Please check all statistical tests performed especially as they relate to reviewer #1 question 9.

We carefully checked the statistics. See the response to reviewer #1 question 9.

4) Please check the comments of review 1 regarding Figure 6--DFP-only group (ie, without the E2)? Does DFP treatment of ApoE mice post OVX prevent the reduced ERalpha expression associated with LMP?

To answer the questions, we remodeled the mice and collected the data. See the responses to reviewer #1 point13 and reviewer #2 point5.

5) Clarification of human data shown in Figure 1 is needed and thorough check of confounders is needed

This information is put in the supplementary data as Table 1. We answered it also in the response to reviewer #2 question 1.

Reviewer #1 (Recommendations for the authors):1) In Figure 1B, can the authors explain why the LPM group has both higher serum iron and tissue iron? Can the authors point to a reference showing higher serum iron in LPM vs EPM? Measure of hepcidin levels would be of interest here as, higher hepcidin levels in the LPM group would explain the high tissue iron/ferritin, but would be expected to lead to lower serum iron levels.

A few references (Pubmed ID: 28620614, 19527179, 11131920 and 1511065) showed higher serum ferritin/iron in LPM though not all point to menopause, but to age up to 70+ years old. We cited the missing references in Revision. Likely, due to being invasive to get the tissues other than only serum, we did not find the tissue data from literatures. In our study though the small number of the samples, the tissue data are luckily available due to the removal of plaques by surgery. Tissue iron and ferritin levels were presented in Figure 1. During revision, serum hepcidin was measured and shown in Figure 1E, explaining the higher tissue iron in LPM group and in agreement with the claim that plasma hepcidin and serum ferritin concentrations are highly correlated (PMID: 19211819). Serum iron is higher in LPM group in our study, likely due to no physiological blood loss, estrogen reduction, and/or the chronic and low inflammation levels in those atherosclerosis patients.

2) In Figure 1E, Ferritin (Ft-L) levels are highly variable among the LPM group. The authors mentioned that the western blot samples in Figure 1E are the same as in figure 1A. However, Ft-L levels measured by IHC in sample P6 and P7 are similar but there are large differences in the Ft-L levels in sample P6 and P7 shown by western blot. Could the authors please explain the reason for this variability across the two techniques.

Sorry for that the sample order in original Figure 1E (Figure 1B in Revision) and 1A. The “same” means the same 8 samples. We realized the order difference in the beginning. Since P6 and P7 are in the same group, we did not redo it. Now we make it clear in the legend.

3) Line 288- Plaque ER expression was lower in LPM as compared to EPM. Please correct the words.

Thank you. We corrected the error.

4) The correlation in Figure 1F is limited by sample size as well as batch effect (ie, the correlation does not exist within the LPM group or within the EPM group?) and should be removed.

We agree. The correlation in original Figure 1F has been removed and we modified the related statements.

5) In Figure 2E in the macrophages, a Western Blot of FPN1 should be included.

Yes, we did. FPN1 has been included in Figure 2E.

6) This sentence needs further clarification: "Ferritin was reduced in response to E2 treatment in the EPM stage (this appears to be only true in macrophages). In contrast, ferritin remained high in the late stage after E2 treatment in both aortae and isolated macrophages (Figure 2D and 2E), which could be explained by the response of Fpn1 expression (clarify this statement) that was decreased in the EPM stage but not in the LPM stage."

This sentence was modified (Line 165-169) as “In the isolated macrophages, ferritin was reduced in EPM group while Fpn1 was elevated in both stages in response to E2 treatment (Figure 2E). These responses were very mild except Fpn1 in early stage (Figure 2D), which suggests that not all cell types are responsive to E2 in aorta tissue, particularly the disrupted iron homeostasis in late stage”.

7) In Figure 3, how do the authors explain the iron-dependent effect of E2 on increasing atherosclerosis in the macrophage FPN KO ApoE KO mice since E2 appears to restore tissue iron and serum iron back towards more normal levels (ie, closer to the ApoE KO mice)? An ApoE + E2 control would also have been helpful here.

Since reviewing editor Q1 repeated this question, we copy from the answer provided above: Physiologically, E2 decreases tissue iron, increases serum iron, and also increases iron uptake to compensate the periodic loss of iron in blood. Then, body iron gradually reach balance and estrogen level goes back to the bottom line. Cellular physiological response to estrogen starts with estrogen binding to either ERa or ERb. ERa is reduced in the macrophage FPN KO Apoe KO mice and administration of E2 may not reach the protective effects through estrogen receptors. However, it has been demonstrated that no protein-bound estrogen has the property to diffuse into cells freely with no regulation and estrogen may have pro- and anti-inflammatory properties depending on the circumstances including different tissues or diseases (PMID: 17640948). Plaques in macrophage FPN KO Apoe KO mice are supposed to challenge more inflammatory situation than the plaques in Apoe KO mice. Therefore, E2 exhibits possible paradox consequences due to timing of E2 administration and iron status.

The effects of E2 on *Apoe*^-/-^ at LPM and EPM stages have been shown in Figure 2, indicating that E2 treatment is beneficial at EPM, but worse at LPM (Figure 2C). Figure 3 is to address the effects of iron and iron±E2. The comparison can be performed between *Apoe*^-/-^ and *Apoe^-/-^Fpn^Lyz2/Lyz2^* and between ± E2 in *Apoe^-/-^Fpn^Lyz2/Lyz2^*. Therefore, *Apoe*^-/-^ +E2 was not set here.

8) In Figure 3, the authors should provide a Western Blot for FPN showing the efficacy of the Cre-Lox model, particularly at the two ages.

Fpn1 KO efficiency was presented in Figure 3 —figure supplement 1A in Revision.

9) The graphs depicted in Figures 4A and 4B have large error bars that intersect, making the conclusion that E2 and iron intersect to regulate ERalpha expression less convincing. Did the authors perform a 2-way ANOVA? What was the p-value for E2? What was the interaction p-value? Clearly, FAC alone decreases ERalpha expression substantially, so why do the authors think it is E2-dependent?

The p-values for E2 were presented in the graphs. The “*” marks had been misplaced in the previous version. Interaction p-value for Figure 4A and B was of no significance. We also performed a two-way ANOVA for Figure 4C and the interaction p-value was 0.0099. The explanation is that ERα was upregulated when cells were treated with E2 alone, but E2+FAC aggravated ERα downregulation compared to FAC only group. So, the interaction p-value for Figure 4A and 4B was seemingly not significant because FAC was treated in all groups, and for Figure 4C definitely significantly. Therefore, we think it is E2-dependent.

10) In Figure 5C, can the authors explain the relative levels compared to the sham group?

We repeated the experiments. The results were consistent. What we think is that FAC addition increases MDM2 expression, promoting ubiquitination and degradation of ERa. Though we see the similar levels of ERa ubiquitination after FAC treatment, part of ubiquitinated ERa could be degraded and not detectable.

11) In Figure 5E, it is unclear why nutlin-3 does not prevent the E2+FAC-mediated reduction of ERalpha?

The efficacy of nutlin-3 in J774a.1 cells seems not as good as in HUVEC. Likely, the sensitivity to nutlin-3 varies in different cell types in response to the interaction of E2 and FAC.

12) For the ApoE experiments, can the authors please provide trends in weight and the food consumptions for each of the mouse groups (for Figures 2, 3, and 6).

Mouse weight was recorded and provided in author response images 1-4, while food consumption was not recorded during the experiments.

**Author response image 1. sa2fig1:** 

**Author response image 3. sa2fig3:** 

**Author response image 4. sa2fig4:** 

13) In Figure 6, do the authors have a DFP-only group (ie, without the E2)? Does DFP treatment of ApoE mice post OVX prevent the reduced ERalpha expression associated with LMP?

In order to add a DFP-only group, we took a half year to remodel the animals. The related data were added to Figure 6. Exactly as reviewer #1 thought, DFP treatment of *Apoe*^-/-^ mice post OVX prevented the reduction of ERalpha expression, consequently greatly benefiting LMP. The efficacy of DFP-only is unexpectedly even better than E2+DFP.

14) In Figure 2, treatment with E2 reduces atherosclerosis in ApoE mice in EPM. In Figure 6, the DFP is administered at the same time as E2. Would pre-treatment with DFP prolong an "EPM state" thus resulting in E2 causing a reduction in atherosclerotic lesions?

This hypothesis is very interesting. It looks like the truth if we just see the DFP-only group. We will work further to test the idea.

Reviewer #2 (Recommendations for the authors):1) The human data as presented in Figure 1 is not convincing given multiple confounders may exist which might affect the data. List patient demographics and compare them directly (age, risk factors, symptomatic, chol lowering meds). What types of plaque were collected? I assume the rationale between late and early is that iron accumulates in late but not early? Make this rationale clear--it is important for the ENTIRE manuscript. Prussian blue is usually blue but I don't see any blue? Higher power please with more details.

Here is the information concerning demographics and comparisons, presented in the new manuscript as Table 1.

Prussian blue is visible when iron level is high enough. Physiologically, we can see some small iron-stained spots in iron-rich tissues, e. g. live and spleen. In other tissues, we generally used DAB enhanced staining to show the relative iron levels. This is the reason why the brown color is seen in Figure 1A. The Method 2.7 gives the details.

2) Mice experiments. Needs to make clear why you looked at early versus late OVX? How was dose of E2 chosen and why. The micrographs of lipid staining are not particularly convincing. It looks as if there is no difference in the late group between saline and E2? Would be useful to show estrogen levels. Why does Fe drop in the late E2 and in Eary E2? Total body iron stores should increase in both groups according to me? Why not show ferritin levels in all mice?

Below we provide answers to all these helpful questions.

Early versus late OVX: we observed the serum and plaque iron difference in LPM and EPM in atherosclerosis patients, more iron in LPM and less iron in EPM. The mouse model mimics the atherosclerosis patients, i. e. early OVX matches to EPM, late OVX to LPM.Dose of E2: 3 μg/kg. This dose is picked by referring to some literatures below: PMID: 30241552 (2 μg/kg), PMID: 30292827 (0.2 mg/kg), PMID: 31141131 (10 μg/kg). Since our experiments last much longer than those, we used a relative low dose of 3 μg/kg.Lipid staining: we assume you mean Figure 2B. Actually, there is a significant difference in the late groups between saline and E2, quantified in Figure 2C. Probably, One of Oil red O-stained aortic lesions in the LPM saline group is misleading. If we count all the lesions in this group versus E2 group, the difference is significant.Serum estrogen levels is presented in Author response image 5 and in Figure 2 —figure supplement 1.E2 treatment increases serum iron, but decreases tissue iron, and promotes iron uptake and release in the small intestine, so total body iron stores are likely increased since the serum hemoglobin is increased post E2 administration (Figure 6 —figure supplement 1). Western blot of ferritin is presented in the manuscript Figure 2 and 3, only not in Figure 6. Instead, we directly measured the tissue iron contents, which is consistent with the data in Figure 2 and 3.

**Author response image 5. sa2fig5:** 

3) In Figure 3 explain more clearly why these timepoints were chosen. You change timepoints from the previous figure why? What is the hypothesis being exposed? All of this is hard to follow.

Possibly, reviewer #2 overlooked our Materials and methods (line 389-). Figure 2 and Figure 3 used different controls. For Figure 2, we chose 8 weeks old as a start, which age is considered as a young adulthood with high expression levels of ERα and serum lipid was not overloaded. OVX was performed, subsequently along with or without E2 treatment to examine the protective effects of estrogen through ERα and the detrimental effects of OVX (loss of estrogen) on iron metabolism. For Figure 3, *Apoe^-/-^FPN1^Lyz2/Lyz2^* mice genetically and gradually got iron overload. Two time points, 16 and 40 weeks, were chosen as adulthood and menopause-age groups. After OVX, the model mimics better the EPM and LPM in human to explore the effect of iron on E2-administration consequence, ERα expression, lesion areas, and more other different perspectives.

The hypothesis in this study is raised based on controversial hormone replacement therapy (HRT) and data presented in Figure 1, as proposed and summarized in Figure 7.

4) IN Figure 4 why are endothelial cells introduced into the experiments?

Endothelial cells are involved in the progress of atherogenesis, especially in plaque angiogenesis. Endothelial cells are also the first line to respond to serum estrogen.

5) In Figure 6 what does Dep do in the absence of OVX? This is a needed control. In addition the amount of Era increase with Dep is very small. Can this really explain the effects of Dep on atherosclerosis? Some verification via other methods is needed.

DFP alone group has been added after a-half-year remodeling. DFP, as an iron chelator, was used for confirming that iron accumulation at LPM stage plays an important role to hinder E2 protective effects through ERa. We provided evidences that the recovery of ERα by iron chelation met the E2 benefits from the development of atherosclerosis. The increased amount of ERa was not very big but significant enough to be observed. The effects of estrogen, as a hormone, is produced via a cascade reaction, thus a significant increase of ERa is capable to contribute to the improvement of AS plaque. The expression of ABCA1 and VEGF could be a proof of ERα increase and its function.

Reviewer #3 (Recommendations for the authors):1. The clinical study involving 20 postmenopausal women is small. It would be important to state somewhere this observation of age-related inverse relationships between iron and ERa levels needs to be replicated in a larger cohort.

Thanks, we added the statement in the Discussion paragraph 2 and 4 and removed the original Figure 1F.

2. It is noted that the study subjects had known atherosclerosis. Do these findings bear out in postmenopausal in general? A comment on this matter seems appropriate.

This is an interesting point, thank you. We added the comment as “Our findings might be applicable to other age and ERα-related postmenopausal diseases, such as osteoporosis.” in Discussion last paragraph (Line 366-368)

3. Line 44-46 in Abstract: "HRT is recommended immediately after menopause along with appropriate iron chelation to protect from atherosclerosis" is too strong based on such small clinical sample and on single study. Please remove or restate.

The wording is reedited as “This study suggests that immediate HRT after menopause along with appropriate iron chelation might provide benefits from atherosclerosis”. (Line 46-47)

4. Line 287-288 (page 10): "By contrast, plaque ERa expression was lower in the EPM than in LPM (Figure 1E)" is wrong. It should be opposite.

Thanks, we corrected.

5. Line 355-358 (page 13): I do not understand what the authors mean by this.

The sentence was modified as “suggesting that other factors may contribute to the aging-related response to estrogen”.

6. Lines 396 (page 14): "massively correlated." Odd word usage. Suggest "strongly correlated"?

Thanks, changed.

7. Line 421 (page 15): "Amazingly" implies the authors were surprised by the result. Suggest "Importantly…".

Modified.

8. Scale back on clinical recommendations. Suggest "this study suggest XYZ." "Additional studies are need to validate these findings…", etc.

The words have been modified accordingly in Abstract and Summary. (Line 46-47, 365-366)

[Editors' note: further revisions were suggested prior to acceptance, as described below.]

The manuscript has been improved but there are some remaining issues that need to be addressed, as outlined below:Reviewer #2 (Recommendations for the authors):The authors have done a good job of responding to my comments and the manuscript is improved and more easy to follow and read.I few remaining comments:Please tone down the abstract: "Thus, iron and estradiol together downregulate ERα through Mdm2-mediated proteolysis, explaining failures of HRT in late postmenopausal subjects with aging-related iron accumulation". The study provides one mechanism that may be important but does not explain the failure of HRT in late post postmenopausal subjects. This is a mouse study.Similarly in the conclusions "This study suggests that immediate HRT after379 menopause along with appropriate iron chelation would provide benefits from380 atherosclerosis." Would implies confirmation--might is a better word.The study demonstrates the role of iron overload in diminishing HRT benefits but it doesn't specifically show a certain mechanism. There certainly need to be some limitations mentioned in the discussion.

Thank you. We agree with this comment and advice. The statements have been toned down appropriately (Line 44-46 and 375) and the limitations are mentioned in Discussion. Specifically, the relevant sentence in the abstract now reads: “Thus, iron and estradiol together downregulate ERα through Mdm2-mediated proteolysis, providing a potential explanation for failures of HRT in late postmenopausal subjects with aging-related iron accumulation.”